# RL³: Boosting Meta Reinforcement Learning via RL inside RL²

## Abstract

Meta reinforcement learning (meta-RL) methods such as RL² have emerged as promising approaches for learning data-efficient RL algorithms tailored to a given task distribution. However, they show poor asymptotic performance and struggle with out-of-distribution tasks because they rely on sequence models, such as recurrent neural networks or transformers, to process experiences rather than summarize them using general-purpose RL components such as value functions. In contrast, traditional RL algorithms are data-inefficient as they do not use domain knowledge, but do converge to an optimal policy in the limit. We propose RL³, a principled hybrid approach that incorporates action-values, learned per task via traditional RL, in the inputs to meta-RL. We show that RL³ earns greater cumulative reward in the long term compared to RL² while drastically reducing meta-training time and generalizes better to out-of-distribution tasks. Experiments are conducted on both custom and benchmark discrete domains from the meta-RL literature that exhibit a range of short-term, long-term, and complex dependencies.

## 1 Introduction

Reinforcement learning (RL) has been shown to produce effective policies in a variety of applications including both virtual (Mnih et al., 2015) and embodied (Schulman et al., 2017; Haarnoja et al., 2018) systems. However, traditional RL algorithms have three major drawbacks: they can be slow to converge, require a large amount of data, and often have difficulty generalizing to out-of-distribution (OOD) tasks not practiced during training. These shortcomings are especially glaring in settings where the goal is to learn policies for a collection or distribution of problems that share some similarities, and for which traditional RL must start from scratch for each problem. For example, many robotic manipulation tasks require interacting with an array of objects with similar but not identical shapes, sizes, weights, materials, and appearances, such as mugs and cups. It is likely that effective manipulation strategies for these tasks will be similar, but they may also differ in ways that make it challenging to learn a single policy that is highly successful on all instances. Recently, meta reinforcement learning (meta-RL) has been proposed as an approach to mitigate these shortcomings by deriving RL algorithms (or meta-RL policies) that *adapt* efficiently to a distribution of tasks that share some common structure (Duan et al., 2016; Wang et al., 2016).

While meta-RL systems represent a significant improvement over traditional RL in such settings, they still require large amounts of data during meta-training time, can have poor asymptotic performance during adaptation, and although they "learn to learn," they often generalize poorly to tasks

Table 1: RL³ combines the strengths of meta-RL (e.g., RL²) and traditional RL. Like RL², RL³ uses finite-context sequence models to represent data-efficient RL algorithms, optimized for tasks within a specified distribution. However, RL³ also includes a general-purpose RL routine that distills arbitrary amounts of data into optimal value-function estimates during adaptation. This improves long-term reasoning and OOD generalization.

| | RL | RL² | RL³ |
|---|:---:|:---:|:---:|
| Short-Term Efficiency | x | ✓ | ✓ |
| Long-Term Performance | ✓ | x | ✓ |
| OOD Generalization | ✓ | x | ✓ |
| | (General Purpose) | | (Improved) |

not represented in the meta-training distribution. This is partly because they rely on black-box sequence models like recurrent neural networks or transformers to process experience data. These models cannot handle arbitrary amounts of data effectively and lack integrated general-purpose RL components that could induce a broader generalization bias.

Hence, we propose $RL^3$, an approach that embeds the strengths of traditional RL within meta-RL. Table 1 highlights our primary aims and the foremost insight informing our approach. The key idea in $RL^3$ is an additional 'object-level' RL procedure executed within the meta-RL architecture that computes task-specific optimal Q-value estimates as supplementary inputs to the meta-learner, in conjunction with sequences of states, actions and rewards. In principle, our approach allows the meta-learner to learn how to optimally fuse raw experience data with summarizations provided by the Q-estimates. Ultimately, $RL^3$ leverages Q-estimates' generality, ability to compress large amounts of experiences into useful summaries, direct actionability, and asymptotic optimality to enhance long-term performance and OOD generalization and drastically reduce meta-training time.

While Q-value estimates can be injected into any other meta-RL algorithm, for clarity of exposition, we implement $RL^3$ by injecting Q-value estimates into one of the most popular and easily understood meta-RL algorithm, $RL^2$ (Duan et al., 2016) (hence, the name $RL^3$). However, it should be noted that *our baseline implementation of $RL^2$ includes significant enhancements* like using transformers instead of LSTMs to improve long-context reasoning, in addition to incorporating numerous recommendations from Ni et al. (2022) that have been shown to make recurrent model-free RL algorithms like $RL^2$ competitive with state-of-the-art meta-RL baselines like VeriBAD (Zintgraf et al., 2020).

The primary contribution of this paper is a proof-of-concept that injecting Q-estimates obtained via traditional object-level RL alongside the typical experience histories within a meta-RL agent leads to higher long-term returns and better OOD generalization, while maintaining short-term efficiency. We further demonstrate that our approach can also work with an abstract, or coarse, representation of the object-level MDP. We experiment with discrete domains that both reflect the challenges faced by meta-RL and simultaneously allow transparent analysis of the results. Finally, we examine the key insights that inform our approach and show theoretically that object-level Q-values are directly related to the optimal meta-value function.

## 2 RELATED WORK

Although meta-RL is a fairly new topic of research, the general concept of meta-learning is decades old (Vilalta & Drissi, 2002), which, coupled with a significant number of design decisions for meta-RL systems, has created a large number of different proposals for how systems ought to best exploit the resources available within their deployment contexts (Beck et al., 2023). At a high level, most meta-RL algorithms can be categorized as either parameterized policy gradient (PPG) models (Finn et al., 2017; Li et al., 2017; Sung et al., 2017; Al-Shedivat et al., 2018; Gupta et al., 2018; Yoon et al., 2018; Stadie et al., 2018; Vuorio et al., 2019; Zintgraf et al., 2019; Raghu et al., 2019; Kaushik et al., 2020; Ghadirzadeh et al., 2021; Mandi et al., 2022) or black box models (Duan et al., 2016; Heess et al., 2015; Wang et al., 2016; Foerster et al., 2018; Mishra et al., 2018; Humplik et al., 2019; Fakoor et al., 2020; Yan et al., 2020; Zintgraf et al., 2020; Liu et al., 2021; Emukpere et al., 2021; Beck et al., 2022). PPG approaches assume that the underlying learning process is best represented as a policy gradient, where the set of parameters that define the underlying algorithm ultimately form a differentiable set of meta-parameters that the meta-RL system may learn to adjust. The additional structure provided by this assumption, combined with the generality of policy gradient methods, means that typically PPG methods retain greater generalization capabilities on out-of-distribution tasks. However, due to their inherent data requirements, PPG methods are often slower to adapt and initially train.

In this paper we focus on black box models, which represent the meta-learning function as a neural network, often a recurrent neural network (RNN) (Duan et al., 2016; Heess et al., 2015; Wang et al., 2016; Humplik et al., 2019; Fakoor et al., 2020; Yan et al., 2020; Zintgraf et al., 2020; Liu et al., 2021) or a transformer (Mishra et al., 2018; Wang et al., 2021; Melo, 2022). There are also several hybrid approaches that combine PPG and black box methods, either during meta-training (Ren et al., 2023) or fine-tuning (Lan et al., 2019; Xiong et al., 2021). Using black box models simplifies the process of augmenting meta states with Q-estimates and allows us to retain relatively better data efficiency while relying on the Q-value injections for better long-term performance and generalization.

Meta-RL systems may also leverage extra information available during training, such as task identification (Humplik et al., 2019; Liu et al., 2021). Such 'privileged information' can of course lead to more performant systems, but is not universally available. As our hypothesis does not rely on the availability of such information, we expect our approach to be orthogonal to, and compatible with, such methods. Black box meta-RL systems that do not use privileged information still vary in several ways, including the choice between on-policy and off-policy learning and, in systems that use neural networks, the choice between transformers (Vaswani et al., 2017) and RNNs (Elman, 1990; Hochreiter & Schmidhuber, 1997; Cho et al., 2014).

The most relevant methods to our work are end-to-end methods, which use a single function approximator to subsume both learner and meta-learner, such as $RL^2$ (Duan et al., 2016), L2L (Wang et al., 2016), SNAIL (Mishra et al., 2018), and E-$RL^2$ (Stadie et al., 2018), and methods that exploit the formal description of the meta-RL problem as a POMDP or a Bayes-adaptive MDP (BAMDP) (Duff, 2002). These methods attempt to learn policies conditioned on the BAMDP belief state while also approximating this belief state by, for example, variational inference (VariBAD) (Zintgraf et al., 2020; Dorfman et al., 2020), or random network distillation on belief states (HyperX) (Zintgraf et al., 2021). Or, they simply encode enough experience history to approximate POMDP beliefs ($RL^2$) (Duan et al., 2016; Wang et al., 2016).

Our proposed method is an end-to-end system that exploits the BAMDP structure of the meta-RL problem by spending a small amount of extra computation to provide inputs to the end-to-end learner that more closely resemble important constituents of BAMDP value functions. Thus, the primary difference between this work and previous work is the injection of Q-value estimates into the meta-RL agent state at each meta-step, in addition to the state-action-reward histories. In this work, our approach, $RL^3$, is implemented by simply injecting Q-value estimates into $RL^2$ alongside experience history, although any other meta-RL algorithm can be used.

## 3 BACKGROUND AND NOTATION

In this section, we briefly cover some notation and concepts upon which this paper is built.

### 3.1 PARTIALLY OBSERVABLE MDPs

We use the standard notation defining a Markov decision process (MDP) as a tuple $M = \langle S, A, T, R \rangle$, where $S$ is a set of states; $A$ is a set of actions; $T$ is the transition and $R$ is the reward function. A *partially observable Markov decision process* (POMDP) extends MDPs to settings with partially observable states. A POMDP is described as a tuple $\langle S, A, T, R, \Omega, O \rangle$, where $S, A, T, R$ are as in an MDP. $\Omega$ is the set of possible observations, and $O : S \times A \times \Omega \to [0, 1]$ is an observation function representing the probability of receiving observation $\omega$ after performing action $a$ and transitioning to state $s'$. POMDPs can alternatively be represented as continuous-state belief-MDPs where a belief state $b \in \Delta^{|S|}$ is a probability distribution over all states. In this representation, a policy $\pi$ is a mapping from belief states to actions, $\pi : \Delta^{|S|} \to A$.

### 3.2 REINFORCEMENT LEARNING

Reinforcement learning (RL) agents learn an optimal policy given an MDP with unknown dynamics using only transition and reward feedback. This is often done by incrementally estimating the optimal action-value function $Q^*(s, a)$ (Watkins & Dayan, 1992), which satisfies the Bellman optimality equation $Q^*(s, a) = \mathbb{E}_{s'}[R(s, a) + \gamma \max_{a' \in A} Q^*(s', a')]$. In large or continuous state settings, it is popular to use deep neural networks to represent the action-value functions (Mnih et al., 2015). We denote the vector representing the Q-estimates of all actions at state $s$ as $Q(s)$, and after $t$ feedback steps, as $Q^t(s)$. Q-learning is known to converge asymptotically (Sutton & Barto, 2018), provided each state-action pair is explored sufficiently. As a rough general statement, $||Q^t(s) - Q^*(s)||_\infty$ is proportional to $\approx \frac{1}{\sqrt{t}}$, with strong results on the convergence error available (Szepesvári, 1997; Kearns & Singh, 1998; Even-Dar et al., 2003). The *theoretical objective* in RL is to optimize the value of the final policy i.e., the cumulative reward per episode, disregarding the data cost incurred and the cumulative reward missed (or *regret*) during learning due to suboptimal exploration.

Figure 1: Overview diagram of RL$^3$ . Black entities represent standard components from RL$^2$, and purple entities represent additions for RL$^3$ . $M_i$ is the current MDP; $s$ is a state; $r$ is a reward; $t_i$ and $t_\tau$ are the amount of time spent experiencing the current MDP and current episode, respectively; $Q_i^t$ is the Q-value estimate for MDP $i$ after $t$ actions; $\nabla \mathcal{J}$ is the policy gradient for meta-training.

## 3.3 META REINFORCEMENT LEARNING

Meta-RL seeks action selection strategies that minimize regret in MDPs drawn from a distribution of MDPs that share the same state and action spaces. Therefore, the objective in meta-RL is to maximize the cumulative reward over the entire interaction (or adaptation) period with an MDP, which may span multiple episodes, in order to optimize the exploration-exploitation tradeoff. Formally,

$$\mathcal{J}(\theta) = \mathbb{E}_{M_i \sim \mathcal{M}} \Big[ \sum_{t=0}^{H} \gamma^t \mathbb{E}_{(s_t, a_t) \sim \rho_i^{\pi_\theta}} [R_i(s_t, a_t)] \Big] \tag{1}$$

where the meta-RL policy $\pi_\theta$ is interpreted as a 'fast' or 'inner' RL algorithm that maps the experience sequence $(s_0, a_0, r_0, ..., s_t)$ within an MDP $M_i$ to an action $a_t$ using either a recurrent neural network or a transformer network. $\rho_i^{\pi_\theta}$ is the state-action occupancy induced by the meta-RL policy in MDP $M_i$, and $H$ is the length of the adaptation period, or *interaction budget*. The objective $\mathcal{J}(\theta)$ is maximized using a conventional 'slow' or 'outer' deep RL algorithm, given the reformulation of the interaction period with an MDP as a single (meta-)episode in the objective function, which maximizes the cumulative reward throughout this period. We will use the term 'experience history', denoted by $\Upsilon$, to refer to the state-action-reward sequence within a meta-episode, which spans across multiple episodes $\{\tau_0, \tau_1, ...\tau_n\}$. Fig. 1 illustrates how these components interconnect.

Another way to conceptualize this problem is to recognize that the meta-RL problem may be written as a meta-level POMDP, where the hidden variable is the particular MDP (or task) at hand, $M_i$, which varies across meta-episodes. This framing, known as Bayesian RL (Ghavamzadeh et al., 2015), leverages the fact that augmenting the task-specific state $s$ with belief over tasks $b(i)$ results in a Markovian meta-state $[s, b]$ for optimal action selection, a model known as the Bayes Adaptive MDP (or BAMDP) (Duff, 2002). That is, this belief state captures all requisite information for the purpose of acting. We will revisit this concept to develop intuition on the role of object-level Q-value estimates in the meta-RL value function.

## 4 RL$^3$

To address the limitations of black box meta-RL methods, we propose RL$^3$, a principled approach that leverages (1) the inherent generality of action-value estimates, (2) their ability to compress experience histories into useful summaries, (3) their direct actionability & asymptotic optimality, (4) their ability to inform task-identification, and (5) their relation to the optimal meta-value function, in order to enhance out-of-distribution (OOD) generalization and performance over extended adaptation periods. The central, novel mechanism in RL$^3$ is an additional 'object-level' RL procedure executed within the meta-RL architecture, shown in Fig. 1, that computes task-specific optimal Q-value estimates $Q_i^t(s_t)$ and state-action counts as supplementary inputs to the meta-RL policy in conjunction with the sequence of states, actions and rewards $(s_0, a_0, r_0, ..., s_t)$. The Q-estimates are computed off-policy, and may involve model estimation and planning, for greater data efficiency. The estimates and the counts are reset at the beginning of each meta-episode as a new task $M_i$ is sampled. In all subsequent text, Q-value estimates used as input entail the inclusion of state-action counts as well. We now present a series of key insights informing our approach.

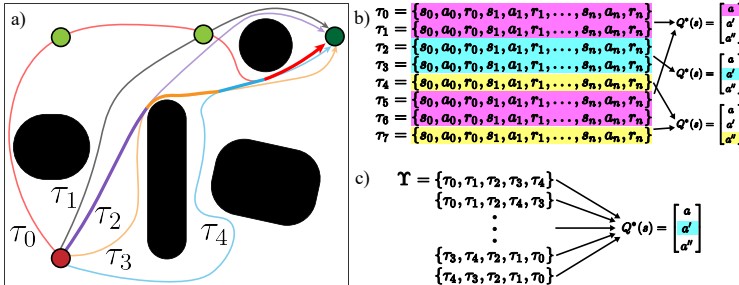

Figure 2: Sub-figure (a) shows a meta-episode in a shortest-path environment where the goal position (green circles) and the obstacles (black regions) may vary across tasks. In this meta-episode, after the meta-RL agent narrows its belief about the goal position of this task (dark-green circle) having followed a principled exploration strategy ($\tau_0$), it explores potential shorter paths in subsequent episodes ($\tau_1, \tau_2, \tau_3, \tau_4$). Throughout this process, the estimated value-function $\hat{Q}^*$ implicitly "remembers" the goal position and previous paths traversed in a finite-size representation, and updates the shortest path calculation (highlighted in bold) using Bellman backups when paths intersect. Sub-figures (b) and (c) illustrate the many-to-one mapping of object- and meta-level data streams to Q-estimates, and thus their utility as compression and summarization mechanisms for meta-learning.

First, estimating action-values is a key component in many **universal** RL algorithms, and asymptotically, they *fully* inform optimal behavior *irrespective of domain*. Strategies for optimal exploration-exploitation trade-off are domain-dependent and rely on historical data, yet many exploration approaches use estimated Q-values and some notion of counts *alone*, such as epsilon-greedy, Boltzmann exploration, upper confidence bounds (UCB/UCT) (Auer, 2002; Kocsis & Szepesvári, 2006), count-based exploration (Tang et al., 2017), curiosity based exploration (Burda et al., 2019) and maximum-entropy RL (Haarnoja et al., 2018). This creates a strong empirical case that using Q-value estimates and state-action counts for efficient exploration has inherent generality.

Second, Q-estimates **summarize experience histories** of arbitrary length *and order* in one constant-size vector. This mapping is many-to-one, and any permutation of transitions ($\langle s, a, r, s' \rangle$ tuples) or episodes in a history of experiences yield the same Q-estimates. Although this compression is lossy, it still "remembers" important aspects of the experienced episodes, such as high-return actions and goal positions (see Fig. 2) since Q-estimates persist across episodes. This simplifies the mapping the meta agent needs to learn as Q-estimates represent a smaller and more salient set of inputs compared to all possible histories with the same implication.

Third, Q-estimates are **actionable**. Estimated off-policy, they explicitly represent the optimal exploitation policy for the current task given the data insofar as the RL module is data-efficient, relieving the meta-RL agent from performing such calculations inside the transformer/RNN. Over time, Q-estimates become more reliable and directly indicate the optimal policy whereas processing raw data becomes more challenging. Fortunately, by incorporating Q-estimates the meta-RL agent can eventually ignore the history in the long run (or towards the end of the interaction period) and simply exploit the Q-estimates by selecting actions greedily.

Fourth, Q-estimates are **excellent task discriminators** and serve as another line of evidence vis-à-vis maintaining belief over tasks. In a simple domain like Bernoulli multi-armed bandits (Duan et al., 2016), Q-estimates and action-counts combined are sufficient for Bayes-optimal behavior even without providing raw experience data – a result surprisingly unstated in the literature to the best of our knowledge (see Appendix A.1). However, Q-estimates and action-counts may not always be sufficient for Bayes-optimal beliefs[1]. In more complex domains, it is hard to prove the sufficiency of Q-estimates regarding task discrimination. However, via empirical analysis in Appendix D, we argue that i) it is highly improbable for two tasks to have similar $Q^*$ functions and ii) Q-estimates tend to become accurate task predictors in just a few steps. This implies that the meta-agent may use this finite summary for task inference rather than relying completely on arbitrarily long histories, potentially contributing to enhanced performance over long adaptation periods.

It can be theoretically argued that since the meta agent is a BAMDP *policy*, it is meta-trained to select greedy actions w.r.t. the BAMDP meta-value function and thus should not require construct-

---

[1] For example, in Gaussian multi-armed bandits, the sufficient statistics include the variance in rewards for each action (see Appendix A.2).

ing a task-specific plan internally. However, the optimality of the meta action-value function depends on implicitly (or explicitly in some approaches (Humplik et al., 2019; Zintgraf et al., 2020; Dorfman et al., 2020; Zintgraf et al., 2021)) maintaining a Bayes-optimal belief over tasks in the transformer/RNN architecture. This may be challenging if the task distribution is too broad and the function approximator is not powerful enough to integrate experience histories into Bayes-optimal beliefs, or altogether impossible if there is a distribution shift at meta-test time. This latter condition is common in practice and is a frequent target use case for meta-RL systems. Incorporating task-specific Q-estimates gives the agent a simple alternative (even if not Bayes-optimal) line of reasoning to translate experiences into actions. Incorporating Q-estimates thus **reduces susceptibility to distribution shifts** since the arguments presented in this section are domain independent.

Finally, Q-estimates often converge far more quickly than the theoretical rate of $\frac{1}{\sqrt{t}}$, allowing them to be useful in the short and medium term, since i) most real-world domains contain significant determinism, ii) it is not necessary to estimate Q-values for states unreachable by the optimal policy, and iii) optimal meta-RL policies may represent active exploration strategies in which Q-estimates converge faster, or evolve in a manner leading to quicker task identification. This is intuitively apparent in shortest-path problems, as illustrated in Fig. 2(a). In a deep neural network, it is difficult to know exactly how Q-estimates will combine with state-action-reward histories when approximating the meta-value function. However, as we show below, we can write an equation for the meta-value function in terms of these constituent streams of information, which may explain why this function is seemingly relatively easy to learn compared to predicting meta-values from histories alone.

### 4.1 THEORETICAL JUSTIFICATION

Here, we consider the interpretation of meta-RL as performing RL on a partially observable Markov decision process (POMDP) in which the partially observable state factor is the identity of the object-level MDP. Without loss of generality, all analysis assumes the infinite horizon setting. We will denote meta-level entities, belonging in this case to a POMDP, with an overbar. For example, we have a meta-level value function $\bar{V}$ and a meta-level belief $\bar{b}$.

First, we show a basic result, that the optimal meta-level value function is upper bounded by the object-level Q-value estimates in the limit.

***Proof:*** Given a task distribution $\mathcal{M}$, then for state $s$, there exists a maximum object-level optimal value function $V^*_{max}(s)$, corresponding to some MDP $M_{max} \in \mathcal{M}$, such that for all MDPs $M_i \in \mathcal{M}, V^*_{max}(s) \geq V^*_i(s)$. The expected cumulative discounted reward experienced by the agent cannot be greater than the most optimistic value function over all tasks, since $\bar{V}^*(\bar{b})$ is a weighted average of individual value functions $V^{\pi_\theta}(s)$, which are themselves upper bounded by $V^*_{max}(s)$. Thus,

$$\max_{M_i \in \mathcal{M}} V^*_i(s) \geq \bar{V}^*(\bar{b}) \quad \forall s \in S. \tag{2}$$

Next, we see that combining the asymptotic accuracy of Q-estimates and Equation equation 2 yields

$$\lim_{t \to \infty} \max_{a \in A, M_i \in \mathcal{M}} Q^t_i(s,a) \geq \bar{V}^*(\bar{b}) \quad \forall s \in S. \quad \square \tag{3}$$

Furthermore, it follows if the meta-level observation $\bar{\omega}$ includes Q-value estimates of the current task $M_i$, it can be shown that as $t \to \infty$, the optimal meta-value function approaches the optimal value function for the current task, i.e., for any $\epsilon > 0$, there exists $\kappa \in \mathbb{N}$ such that for $t \geq \kappa$,

$$\left| \max_{a \in A} \left[ Q^t_i(s,a) \right] - \bar{V}^*(\bar{b}) \right| \leq \epsilon \quad \forall s \in S. \tag{4}$$

Equation 4 (proof in Appendix A.3) shows that for $t \geq \kappa$, acting greedily w.r.t. $Q^*_i$ leads to Bayes-optimal behavior, and knowing the Bayes-optimal belief over tasks is not required, implying that the experience history can be ignored at that point. Moreover, it follows from equation 4 that for $t < \kappa$,

$$\bar{V}^*(\bar{b}) = \max_{a \in A} \left[ Q^t_i(s,a) \right] + \varepsilon_i(\Upsilon) \tag{5}$$

where error $\varepsilon_i(\Upsilon)$ is the error in Q-value estimates. While this error will diminish as $t \to \infty$, in the short run, a function $f(\Upsilon)$ could be learned to either estimate the error or estimate $\bar{V}^*(\bar{b})$ entirely.

The better performance of RL³ could be explained by either error $\varepsilon_i(\Upsilon)$ being simpler to estimate, or, the meta-agent behavior being more robust to errors in estimates of $\varepsilon_i(\Upsilon)$ when Q-estimates are

supplied directly as inputs, than to errors in a more complicated approximation of $\bar{V}^*(\bar{b})$. Moreover, this composition benefits from the fact that the convergence rate for Q-estimates suggests a natural, predictable rate of shifting reliance from $f(\Upsilon)$ to $Q_i^t(s)$ as $t \to \infty$. However, we do not bake this structure into the network and instead let it implicitly learn how much to use the Q-estimates.

Finally, we note that near-perfect function approximation of $\bar{V}^*(\bar{b})$ as $t \to \infty$ reduces error in meta-value function approximation for all preceding belief states, as meta-values for consecutive belief states $\bar{b}$ and $\bar{b}'$ are linked through the Bellman equation for BAMDPs (see details in Appendix A.3)

$$\bar{V}^*(\bar{b}) = \max_{a \in A} \Big[ \sum_{M_i \in \mathcal{M}} \bar{b}(i) R_i(s, a) + \gamma \sum_{\bar{\omega} \in \bar{\Omega}} \bar{O}(\bar{\omega}|\bar{b}, a) \bar{V}^*(\bar{b}') \Big]. \tag{6}$$

This dependency helps meta-training in RL$^3$ with temporal-difference based learning algorithms. Without conditioning on Q-estimates, error in $\bar{V}^*(\bar{b})$ would instead increase as $t \to \infty$, as the meta-critic would be conditioned on a larger history, which could destabilize the meta-value learning for all preceding belief states during meta-training.

## 4.2 IMPLEMENTATION

Implementing RL$^3$ involves simply replacing each MDP in the task distribution with a corresponding value-augmented MDP (VAMDP) and solving the resulting VAMDP distribution using RL$^2$. Each VAMDP has the same action space and reward function as the corresponding MDP. The value aug-mented state $\hat{s}_t \in S \times \mathbb{R}^k \times \mathbb{I}^k$ includes the object level state $s_t$, $k$ real values and $k$ integer values for the Q-estimates $(Q^t(s_t, a))$ and action counts $(N^t(s_t, a))$ for each of the $k$ actions. In practice, we provide action advantages along with the max Q-value (value function) instead of Q-estimates. When the object-level state space $S$ is discrete, $s_t$ needs to be represented as an $|S|$-dimensional one-hot vector. Note that the value augmented state space is continuous. In the VAMDP transition function, the object-level state $s$ has the same transition dynamics as the original MDP, while the dynamics of Q-estimates are a function of $T$, $R$, and the specific object-level RL algorithm used for estimating Q-values. An episode of the VAMDP spans the entire interaction period with the corre-sponding MDP, which may include multiple episodes of the MDP, as Q-estimates continue to evolve beyond episode boundaries. In code, a VAMDP RL environment is implemented as a wrapper over a given MDP environment. The pseudocode, additional implementation details and hyperparameters for RL$^2$ and RL$^3$ are mentioned in Appendix B.

## 5 EXPERIMENTS

We compare RL$^3$ to our enhanced implementation of RL$^2$. In our implementation, we replace LSTMs with transformers in both the meta-actor and meta-critic for the purpose of mapping ex-periences to actions and meta-values, respectively. This is done to improve RL$^2$'s ability to handle long-term dependencies instead of suffering from vanishing gradients. Moreover, RL$^2$-transformer trains significantly faster than RL$^2$-LSTM. Second, we include in the state space the total number of interaction steps and the total number of steps within each episode during a meta-episode (see Fig. 1). Third, we use PPO (Schulman et al., 2017) for training the meta actor-critic, instead of TRPO (Schulman et al., 2015). These modifications and other minor-implementation details incor-porate the recommendations made by Ni et al. (2022), who show that model-free recurrent RL is competitive with other state-of-the-art meta RL approaches such as VeriBAD (Zintgraf et al., 2020), if implemented properly. RL$^3$ simply applies the modified version of RL$^2$ to the distribution of value-augmented MDPs explained in section 4.2. Within each VAMDP, our choice of object-level RL is a model-based algorithm to maximize data efficiency – we estimate a tabular model of the environment and run finite-horizon value-iteration using the model. Once again, we emphasize that the core of our approach, which is augmenting MDP states with action-value estimates, is not inher-ently tied to RL$^2$ and is orthogonal to most other meta-RL research. VAMDPs can be plugged into any base meta-RL algorithm with a reasonable expectation of improving it.

In our test domains, each meta-episode involves procedurally generating an MDP according to a parameterized distribution, which the meta-actor interacts with for a fixed adaptation period, or interaction budget, $H$. This interaction might consist of multiple object-level episodes of variable length, each of which are no longer than a maximum task horizon. For a given experiment, each

Table 2: Test scores (mean $\pm$ standard error) for Bandits domain and the $^\dagger$OOD variation.

| Budget $H$ | $\text{RL}^2$ | $\text{RL}^3$ | $\text{RL}^3$ (Markov) |
|---|---|---|---|
| 100 | $76.9 \pm 0.6$ | $77.5 \pm 0.5$ | $75.2 \pm 0.5$ |
| 500 | $392.1 \pm 2.5$ | $393.2 \pm 2.7$ | $391.75 \pm 2.6$ |
| $500^\dagger$ | $430.2 \pm 2.8$ | $\mathbf{434.9} \pm 2.8$ | $433.7 \pm 2.8$ |

approach is trained on the same series of MDPs. Each experiment is done for 3 seeds and the results of the median performing model are reported. For testing, each approach is evaluated on an identical set of 1000 MDPs distinct from the training MDPs. For testing OOD generalization, MDPs are generated from distributions with different parameters than in training. We select three discrete domains for our experiments, which cover a range of short-term, long-term, and complex dependencies. These domains both reflect the challenges faced by meta-RL and simultaneously allow transparent analysis of the results.

**Bernoulli Bandits**: We use the same setup described by Duan et al. (2016) with $k = 5$ arms. To test OOD generalization, we generate bandit tasks by sampling success probabilities from $\mathcal{N}(0.5, 0.5)$. We should note that this is an easy domain and serves as a sanity check to ensure that Q-value estimates do not hurt $\text{RL}^3$, causing inferior performance.

**Random MDPs:** We use the same setup described by Duan et al. (2016). The MDPs have 10 states, 5 actions, and task horizon 10. The rewards and transition probabilities are drawn from a normal and a flat Dirichlet distribution ($\alpha = 1.0$), respectively. OOD test MDPs use Dirichlet $\alpha = 0.25$. We should note that this domain is particularly challenging for $\text{RL}^3$ due to the high degree of stochasticity and thus the slower convergence rate of Q-estimates.

**GridWorld Navigation:** A set of navigation tasks in a 2D grid environment. We experiment with 11x11 (121 states) and 13x13 (169 states) grids. The agent starts in the center and needs to navigate through obstacles to a single goal. The grid also contains slippery tiles, dangerous tiles and warning tiles. See Fig. 4(a) for an example of a 13x13 grid. The state representation is coordinates $(x, y)$. To test OOD generalization, we vary parameters including the stochasticity of actions, density of obstacles and the number of dangerous tiles. For this domain, we consider an additional variation of $\text{RL}^3$, called $\text{RL}^3$-*coarse* where a given grid is partitioned into clusters of 2 adjacent tiles (or abstract states), which are used *solely* for the purpose of estimating the object-level Q-values. Our goal is to test whether coarse-level Q-value estimates are still useful to the meta-RL policy. The domains and the abstraction strategy are described in greater detail in Appendices E and B.3, respectively.

# 6 RESULTS

In summary, we observe that beyond matching or exceeding the performance of $\text{RL}^2$ in all test domains i) $\text{RL}^3$ shows better OOD generalization, which we attribute to the increased generality of the Q-value representation, ii) the advantages of $\text{RL}^3$ increase with longer interactions periods and less stochastic tasks, which we attribute to the increased accuracy of the Q-value estimates, iii) $\text{RL}^3$ performs well even with coarse-grained object-level RL over abstract states with substantial computational savings, showing minimal drop in performance in most cases, and iv) $\text{RL}^3$ shows faster meta-training.

**Bandits:** Fig 2 shows the results for this sanity-check domain. For $H = 100$ and $H = 500$, both approaches perform comparably. However, the OOD generalization for $\text{RL}^3$ is slightly better. We also experiment with a Markovian version of $\text{RL}^3$, where a feed-forward neural network is conditioned only on the Q-estimates and action-counts, since those are sufficient for Bayes-optimal behavior in this domain. As expected, the results are similar to regular $\text{RL}^3$ .

**MDPs:** Figures 3a and 3b show the results for the MDPs domain. In Figure 3a, we see that for relatively short budgets, $H \leq 500$, both $\text{RL}^3$ and $\text{RL}^2$-transformer perform comparably on in-distribution problems, with $\text{RL}^3$ performing slightly better on OOD tasks. We suspect that, due to the short budgets and highly stochastic domain, Q-estimates do not converge enough to be very useful for $\text{RL}^3$ . However, as the budget increases, we see that $\text{RL}^3$ continues to improve while $\text{RL}^2$-transformer actually becomes worse and the performance gap on both in-distribution and OOD tasks becomes significant. Overall, we see that $\text{RL}^3$ *preserves asymptotic scaling properties of traditional RL while simultaneously maintaining strong OOD performance.* Moreover $\text{RL}^3$ it is able to

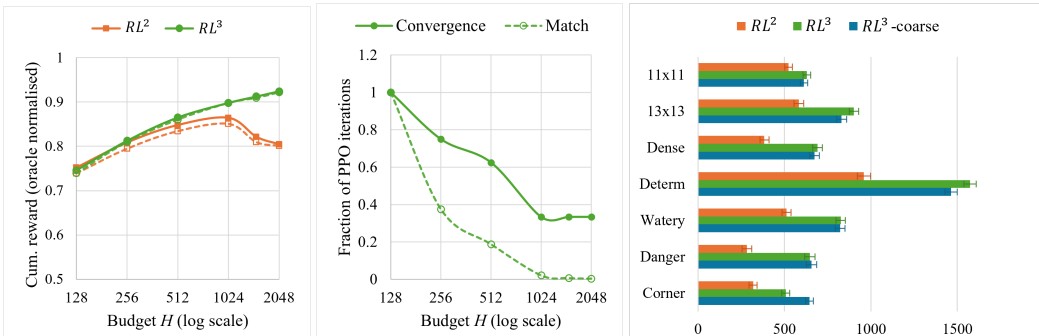

Figure 3: Results for the MDPs and GridWorlds domains. Figure 3a shows the average cumulative reward (negligible standard error) earned as a fraction of the oracle policy for in-distribution (solid) and OOD (dashed) tasks; Figure 3b shows the fraction of $RL^2$-transformer meta-training iterations that $RL^3$ requires (variance is insignificant across seeds) to match $RL^2$-transformer performance or fully converge, both as functions of the adaptation period. Note the log horizontal axis on both plots. Figure 3c shows the average cumulative reward ($\pm$ standard error) earned by $RL^2$, $RL^3$, and $RL^3$-coarse agents on several variations of the GridWorlds domain.

learn meta-policies much more efficiently. Figure 3b shows the number of iterations of PPO $RL^3$ takes to converge completely, as well as to match the performance of $RL^2$-transformer, measured as a fraction of the time it takes for $RL^2$-transformer to converge. This advantage of $RL^3$ is again most pronounced for longer adaptation periods, but we still do observe significant meta-training speedup on even moderate ones. Overall, it is clear that as adaptation periods grow, $RL^3$ achieves nearer-to-optimal policies in a fraction of the meta-training time and maintains better OOD generalization.

**GridWorlds:** Fig 3c shows the results for the GridWorld domain. On 11x11 grids with $H = 250$, $RL^3$ significantly outperforms $RL^2$. On 13x13 grids with $H = 350$, the performance margin is even greater, showing that while $RL^2$-transformer struggles with a greater number of states, a longer adaptation period and more long-term dependencies, $RL^3$ can take advantage of the Q-estimates to overcome the challenge. We also test the OOD generalization of both approaches in different ways by varying certain parameters of the 13x13 grids, namely, increasing the obstacle density (DENSE), making actions on non-water tiles deterministic (DETERMINISTIC), increasing the number of wet 'W' tiles (WATERY), increasing the number of danger 'X' tiles (DANGEROUS) and having the goal only in the corners (CORNER). On all variations, $RL^3$ continues to significantly outperform $RL^2$. In a particularly interesting outcome, both approaches show improved performance on the DETERMINIS-TIC variation. However, $RL^3$ gains 80% more points than $RL^2$, which is likely because Q-estimates converge faster on this less stochastic MDP and therefore provide greater help to $RL^3$. Conversely, in the WATERY variation, which is more stochastic, both $RL^2$ and $RL^3$ lose roughly equal number of points. Overall, in each case, $RL^3$-coarse significantly outperforms $RL^2$-transformer. In fact, it performs on par with $RL^3$, even outperforming it on CORNER variation, except on the canonical 13x13 case and its DETERMINISTIC variation, where it scores about 90% of the scores for $RL^3$. Finally, we see similar meta-training speedups where $RL^3$ requires just 50% and 30% of the total iterations to match the performance of $RL^2$-transformer on the 11x11 and 13x13 grids, respectively.

Fig. 4 shows a sequence of snapshots of a meta-episode where the trained $RL^3$ agent is interacting with an instance of a 13x13 grid. The first snapshot shows the agent just before reaching the goal for the first time. Prior to the first snapshot, the agent had explored many locations in the grid. The second snapshot shows the next episode just after the agent finds the goal, resulting in value estimates being updated using object-level RL for all visited states. Snapshot 3 shows the agent consequently using the Q-estimates to navigate to the goal presumably by choosing high-value actions. The agent also explores several new nearby states for which it does not have Q-estimates. Snapshot 4 shows the final Q-value estimates. A set of short videos of the GridWorld environment, showing both $RL^2$ and $RL^3$ agents solving the same set of problem instances, is included in the supplementary material.

**Computation Overhead Considerations:** As mentioned earlier, for implementing object-level RL, we use model estimation followed by finite-horizon value-iteration to obtain Q-estimates. The computation overhead is negligible for Bandits (5 actions, task horizon = 1) and very little for the MDPs domain (10 states, 5 actions, task horizon 10). For 13x13 GridWorlds (up to 169 states, 5 actions, task horizon = 350), $RL^3$ takes approximately twice the computation time of $RL^2$ per meta-episode. However, $RL^3$-coarse requires only 10% overhead while still outperforming $RL^2$ and retaining more

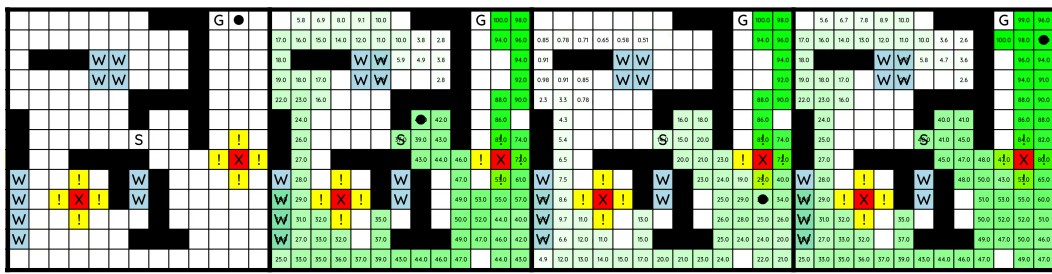

Figure 4: An RL$^3$ policy on a selected meta-episode visualized using a sequence of snapshots. 'S' is the starting tile, 'G' is the goal tile and the black circle shows the current position of the agent. Blue tiles marked 'W' are wet tiles. Wet tiles always lead to the agent slipping to one of the directions orthogonal to the intended direction of movement. Entering wet tiles yield an immediate reward of -2. Yellow tiles marked '!' are warning tiles and entering them causes -10 reward. Red tiles marked 'X' are fatally dangerous. Entering them ends the episode and leads to a reward of -100. Black tiles are obstacles. White tiles yield a reward of -1 to incentive the agent to reach the goal quickly. On all tiles other than wet tiles, there is a chance of slipping sideways with a probability of 0.2. The object-level state-values $v^t(s) = \max_a Q^t(s, a)$, as approximated by object-level RL, is represented using shades of green (and the accompanying text), where darker shades represent higher values.

than 90% of the performance of RL$^3$. This demonstrates the utility of state abstractions in RL$^3$ for scaling. Finally, the meta-training sample efficiency demonstrated by RL$^3$ translates directly to wall-time efficiency as training is dominated by gradient computation, *not* value iteration during data collection in PPO. Our implementation is available in the supplementary material.

# 7 LIMITATIONS AND CONCLUSION

Though it compares favorably to strong meta-RL approaches like RL$^2$-transformer where applicable, RL$^3$ does have some limitations. First, it assumes the object-level decision-making model is an MDP, which although a common assumption in the literature, may be challenged in practice. While in principle we could extend RL$^3$ to POMDPs using methods like point-based value iteration, this has yet to be tested empirically. Second, RL$^3$ relies on fast, potentially approximate methods for object-level RL, and using value iteration complicates application to problems with continuous state spaces. However, we speculate that a crude linear function approximation would suffice. Finally, inference time is slightly slower at deploy time due to running object-level RL. However, the overall training time is actually faster because of better meta-training efficiency. In fact, RL$^3$ could enable working with adaptation periods that are otherwise prohibitively long for many meta-RL approaches.

To conclude, in this paper, we introduced RL$^3$, a principled hybrid approach that combines the strengths of traditional RL and meta-RL and provides a more robust and efficient meta-RL algorithm. We advanced intuitive and theoretical arguments regarding its suitability for meta-RL and presented empirical evidence to validate those ideas. Specifically, we demonstrated that RL$^3$ holds potential to enhance long-term performance, generalization on out-of-distribution tasks and reducing meta-training time. In future work, we plan to explore extending RL$^3$ to handle continuous state spaces.

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

# A PROOFS

## A.1 BAYES OPTIMALITY OF Q-VALUE ESTIMATES IN BERNOULLI MULTI-ARMED BANDITS

Given an instance of a Bernoulli multi-armed bandit MDP, $M_i \sim \mathcal{M}$, and trajectory data $\Upsilon_{1:T}$ up to time $T$, we would like to show that the probability $P(i|\Upsilon_{1:T})$ can be determined entirely from $Q$-estimates $Q_i^T$ and action-counts $N_i^T$, as long as the initial belief is uniform or known.

In the following proof, we represent an instance $i$ of $K$-armed Bandits as a $K$-dimensional vector of success probabilities $[p_{i1}, ..., p_{iK}]$, such that pulling arm $k$ is associated with reward distribution $P(r = 1|i, k) = p_{ik}$ and $P(r = 0|i, k) = (1 - p_{ik})$.

Let the number of times arm $k$ is pulled up to time $T$ be $N_{ik}^T$, and the number of successes associated with pulling arm $k$ up to time $T$ be $q_{ik}^T$. Given that this is an MDP with just a single state and task horizon of 1, the $Q$-estimate associated with arm $k$ is just the average reward for that action, which is the ratio of successes to counts associated with that action i.e., $Q_{ik}^T = \frac{q_{ik}^T}{N_{ik}^T}$. To reduce the clutter in the notation, we will drop the superscript $T$ for the rest of the subsection.

Now,

$$P(i|\Upsilon_{1:T}) = \alpha P(i) \cdot P(\Upsilon_{1:T}|i) \tag{7}$$

where $\alpha$ is the normalization constant, $P(i)$ is the prior probability of task $i$ (which is assumed to be known beforehand), and $\Upsilon_{1:T}$ is the sequence of actions and the corresponding rewards up to time $T$. Assuming, without loss of generality, that the sequence of actions used to disambiguate tasks is a given, $P(\Upsilon_{1:T}|i)$ becomes simply the product of probabilities of reward outcomes up to time $T$, noting that the events are independent. Therefore,

$$P(\Upsilon_{1:T}|i) = \prod_{k=1:K} \prod_{t=1:T} ([r_{tk} = 1]p_{ik} + [r_{tk} = 0](1 - p_{ik})) \tag{8}$$

$$= \prod_{k=1:K} p_{ik}^{q_{ik}} \cdot (1 - p_{ik})^{N_{ik} - q_{ik}} \tag{9}$$

$$= \prod_{k=1:K} p_{ik}^{Q_{ik} N_{ik}} \cdot (1 - p_{ik})^{N_{ik} - Q_{ik} N_{ik}} \tag{10}$$

Putting everything together,

$$P(i|\Upsilon_{1:T}) = \alpha P(i) \cdot \prod_{k=1:K} p_{ik}^{Q_{ik} N_{ik}} \cdot (1 - p_{ik})^{N_{ik} - Q_{ik} N_{ik}} \tag{11}$$

This equation proves that $N_i^T$ and $Q_i^T$ are sufficient statistics to determine $P(i|\Upsilon_{1:T})$ in this domain, assuming that the prior over task distribution is known. $\square$

## A.2 NON-BAYES OPTIMALITY OF Q-VALUE ESTIMATES IN GAUSSIAN MULTI-ARMED BANDITS

Given an instance of a Gaussian multi-armed bandit MDP, $M_i \sim \mathcal{M}$, and trajectory data $\Upsilon_{1:T}$ up to time $t$, here we derive the closed-form expression of the probability $P(i|\Upsilon_{1:T})$ and show that it contains terms other than $Q$-estimates $Q_i^t$ and action-counts $N_i^t$.

In the following proof, we represent an instance $i$ of $K$-armed Bandits as a $2K$-dimensional vector of means and standard deviations $[\mu_{i1}, ..., \mu_{iK}, \sigma_{i1}, ..., \sigma_{iK}]$, such that pulling arm $k$ is associated with reward distribution $P(r|i, k) = \frac{1}{\sqrt{2\pi}\sigma_{ik}} \exp(\frac{r - \mu_{ik}}{\sigma_{ik}})^2$.

Let the number of times arm $k$ is pulled up to time $T$ be $N_{ik}^T$. Given that this is an MDP with just a single state and the task horizon is 1, the $Q$-estimate associated with arm $k$ is just the average

reward for that action $\text{Avg}[r_k]$ up to time $T$. To reduce the clutter in the notation, we will drop the superscript $T$ for the rest of the subsection.

As in the previous subsection, we now compute the likelihood $P(\Upsilon_{1:T}|i)$.

$$P(\Upsilon_{1:T}|i) = \prod_{k=1:K} \prod_{t=1:T} \frac{1}{\sqrt{2\pi}\sigma_{ik}} \exp(\frac{r_{tk} - \mu_{ik}}{\sigma_{ik}})^2 \tag{12}$$

Therefore, the log likelihood is

$$\log P(\Upsilon_{1:T}|i) = \sum_{k=1:K} \sum_{t=1:T} \frac{(r_{tk} - \mu_{ik})^2}{\sigma_{ik}^2} - \log(2\pi\sigma_{ik})/2 \tag{13}$$

$$= \sum_{k=1:K} N_{ik} \frac{\text{Avg}[(r_{tk} - \mu_{ik})^2]}{\sigma_{ik}^2}$$

$$- N_{ik} \log(2\pi\sigma_{ik})/2 \tag{14}$$

$$= \sum_{k=1:K} N_{ik} \frac{\text{Avg}[r_k^2] - 2\mu_{ik}\text{Avg}[r_k] + \mu_{ik}^2}{\sigma_{ik}^2}$$

$$- N_{ik} \log(2\pi\sigma_{ik})/2 \tag{15}$$

$$= \sum_{k=1:K} N_{ik} \frac{(\text{Var}[r_k] + \text{Avg}[r_k]^2) - 2\mu_{ik}\text{Avg}[r_k] + \mu_{ik}^2}{\sigma_{ik}^2}$$

$$- N_{ik} \log(2\pi\sigma_{ik})/2 \tag{16}$$

$$= \sum_{k=1:K} N_{ik} \frac{\text{Var}[r_k] + (Q_{ik})^2 - 2\mu_{ik}Q_{ik} + \mu_{ik}^2}{\sigma_{ik}^2}$$

$$- N_{ik} \log(2\pi\sigma_{ik})/2 \tag{17}$$

Therefore, computing this expression requires computing the variance in rewards, $\text{Var}[r_k]$, associated with each arm up to time $T$, apart from the $Q$-estimates and action-counts. This proves that $Q$-estimates and action-counts alone are insufficient to completely determine $P(i|\Upsilon_{1:T})$ in Gaussian multi-armed bandits domain. $\qquad\square$

### A.3 OBJECT-LEVEL Q-ESTIMATES AND META-LEVEL VALUES

***Proof of Equation 4:*** In standard meta-RL, the only observed variable in the POMDP state $\bar{s}_t = [s_t, i]$ at time $t$ is the state $s_t$ of the current MDP i.e., $\bar{\omega}_t = s_t$, while the task identity $i$ is hidden. However, in RL$^3$, $\bar{\omega}_t$ includes the vector of $Q$-estimates $Q_i^t(s_t)$ for the hidden task, which means that the meta-level observation function $\bar{O}(\bar{\omega}|\bar{b}, a)$ factors in the probability that a particular $Q$-esimate will be observed following an action $a$ given an initial belief $\bar{b}$ state. (Note that we will use $\bar{b}(\bar{s})$ and $\bar{b}(i)$ interchangeably since $i$ is the only hidden variable in $\bar{s}$). In practice, such $Q$-value estimates provide excellent evidence (see Appendix D) for task identification. This allows for robust belief recovery even if the initial belief is not Bayes-optimal (or altogether not maintained), especially as the Q-estimates converge and stabilize in the limit, leading to two cases:

**Case 1:** The observed $Q$-values are unique to MDP $M_i$. In this case, the belief distribution will collapse rapidly to zero for tasks $j \neq i$, and thus $\max_{a \in A} Q_i(s, a) = \bar{V}^*(\bar{b})$.

**Case 2:** The observed $Q$-values are not unique. In this case, belief will not collapse to a single MDP. However, belief will still reduce to zero for tasks not compatible with the observed $Q$-values. The meta-level value function $\bar{V}^*(\bar{b})$, which will be an expectation over object-level values, will simplify to $\max_{a \in A} Q_i(s, a)$ since $Q$-values for all remaining tasks are identical, where $i$ may represent any of the (identical $Q$-valued) tasks with non-zero belief.

This proves equation 4. Note that in the limit, the task can be identified perfectly from the stream of experiences as all state-action pairs are explored, and the meta-level value function becomes equivalent to the optimal object-level value function of the identified (or current) task. However, the above proof demonstrates that RL$^3$ can infer this equivalency implicitly in the limit without relying on the stream of experiences or identifying the task fully, and furthermore, directly model the meta-value function in terms of the supplied object-level value function. $\qquad\square$

***Proof of Equation 6:*** We first write the Bellman equation for the optimal meta-level POMDP value function in its belief-MDP representation:

$$\bar{V}^*(\bar{b}) = \max_{a \in A} \Big[ \sum_{\bar{s} \in \bar{S}} \bar{b}(\bar{s}) \bar{R}(\bar{s}, a) + \gamma \sum_{\bar{\omega} \in \bar{\Omega}} \bar{O}(\bar{\omega}|\bar{b}, a) \bar{V}^*(\bar{b}') \Big]. \tag{18}$$

However, given that in the POMDP state $\bar{s} = [s, i]$, the only hidden variable is the task $i$, we can re-write this as

$$\bar{V}^*(\bar{b}) = \max_{a \in A} \Big[ \sum_{M_i \in \mathcal{M}} \bar{b}(i) R_i(s, a) + \gamma \sum_{\bar{\omega} \in \bar{\Omega}} \bar{O}(\bar{\omega}|\bar{b}, a) \bar{V}^*(\bar{b}') \Big], \tag{19}$$

where $\bar{b}(i)$ denotes the meta-level belief that the agent is operating in MDP $M_i$, and $R_i(s, a)$ is the reward experienced by the agent if it executes action $a$ in state $s$ in MDP $M_i$. Here, $\bar{b}'$ may be calculated via the belief update as in §3.1. $\qquad\square$

## B  ARCHITECTURE

### B.1  RL$^2$

Our modified implementation of RL$^2$ uses transformer decoders (Vaswani et al., 2017) instead of RNNs to map trajectories to action probabilities and meta-values, in the actor and the critic, respectively, and uses PPO instead of TRPO for outer RL. The decoder architecture is similar to (Vaswani et al., 2017), with 2 layers of masked multi-headed attention. However, we use learned position embeddings instead of sinusoidal, followed by layer normalization. Our overall setup is similar to (Esslinger et al., 2022).

For each meta-episode of interactions with an MDP $M_i$, the actor and the critic transformers look at the entire history of experiences up to time $t$ and output the corresponding action probabilities $\pi_1 ... \pi_t$ and meta-values $\bar{V}_1 ... \bar{V}_t$, respectively. An experience input to the transformer at time $t$ consists of the previous action $a_{t-1}$, the latest reward $r_{t-1}$, the current state $s_t$, episode time step $t_\tau$, and the meta-episode time step $t$, all of which are normalized to be in the range $[0, 1]$. In order to reduce inference complexity, say at time step $t$, we append $t$ new attention scores (corresponding to experience input $t$ w.r.t. the previous $t-1$ experience inputs) to a previously cached $(t-1) \times (t-1)$ attention matrix, instead of recomputing the entire $t \times t$ attention matrix. This caching mechanism is implemented for each attention head and reduces the inference complexity at time $t$ from $\mathcal{O}(t^2)$ to $\mathcal{O}(t)$.

### B.2  RL$^3$

The input of the transformer in RL$^3$ includes a vector of $Q$ estimates (in practice, they are supplied as the vector of advantage estimates $(Q - \max_a Q)$ along with the value function $(\max_a Q)$ separately) and a vector of action counts at each step $t$ for the corresponding state. As mentioned in Section 4.2, this is implemented in our code simply by converting MDPs in the problem set to VAMDPs using a wrapper and running our implementation of RL$^2$ thereafter. The pseudocode is shown in the algorithm 1. The Markov version of RL$^3$ uses a dense neural network, with two hidden layers of 64 nodes each, with the ReLU activation function.

For object-level RL, we use model estimation followed by value iteration (with discount factor $\gamma = 1$) to obtain $Q$-estimates. The transition probabilities and the mean rewards are estimated using maximum likelihood estimation (MLE), with Laplace smoothing (coefficient = 0.1) for transition probabilities estimation. For unseen actions, rewards are assumed to be zero, and transitions equally likely to other states. States are added to the model incrementally when they are visited, so that value iteration does not compute values for unvisited states. Moreover, value iteration is carried out only for iterations equal to the task horizon (which is 1, 10, 250, 350 for Bandits, MDPs, 11x11 GridWorld, 13x13 GridWorld domains, respectively), unless the maximum Bellman error drops below 0.01.

---

**Algorithm 1** Value-Augmenting Wrapper for Discrete MDPs

---

   **procedure** RESETMDP(vamdp)
       vamdp.$t \leftarrow 0$; vamdp.$t_\tau \leftarrow 0$
       vamdp.$N[s, a] \leftarrow 0$; vamdp.$Q[s, a] \leftarrow 0 \quad \forall s \in S, a \in A$
       vamdp.rl $\leftarrow$ INITRL()
       $s =$ RESETMDP(vamdp.mdp)
       **return** ONEHOT($s$) $\cdot Q[s] \cdot N[s]$
   **procedure** STEPMDP(vamdp, $a$)
       $s \leftarrow$ mdp.$s$
       $r, s' \leftarrow$ STEPMDP(vamdp.mdp, $a$)
       $d \leftarrow$ TERMINATED(vamdp.mdp)
       vamdp.$t$, vamdp.$N[s, a]$, vamdp.$t_\tau \leftarrow$ += 1
       vamdp.$Q \leftarrow$ UPDATERL(vamdp.rl, $s, a, r, s', d$)
       **if** $d$ **or** vamdp.$t_\tau \geq$ task_horizon **then**
          vamdp.$t_\tau \leftarrow 0$
          $s' \leftarrow$ RESETMDP(vamdp.mdp)
       **return** $r$, ONEHOT($s'$) $\cdot Q[s'] \cdot N[s']$       ▷ Concatenate state, $Q$-estimates and action counts
   **procedure** TERMINATED(vamdp)
       **return** vamdp.$t \geq H$

---

### B.3 RL³-COARSE

During model estimation in RL³-coarse, concrete states in the underlying MDP are incrementally clustered into abstract states as they are visited. When a new concrete state is encountered, its abstract state ID is set to that of a previously visited state within a 'clustering radius', unless that previous state is already part of a full cluster (determined by a maximum 'cluster size' parameter). If multiple visited states satisfy the criteria, the ID of the closet one is chosen. If none of the visited states that satisfy the criteria, then the new state is assigned a new abstract state ID, increasing the number of abstract states in the model. It is worth noting that this method of deriving abstractions does not take advantage of any structure in the underlying domain. However, this simplicity makes it general purpose, efficient, and impartial, while still leading to excellent performance. For our GridWorld domain, we chose a cluster size of 2 and a clustering radius such that only non-diagonal adjacent states are clustered (Manhattan radius of 1).

The mechanism for learning the transition function and the reward function in the abstract MDP is the same as before. For estimating $Q$-values for a given concrete state, value iteration is carried out on the abstract MDP and the $Q$-estimates of the corresponding abstract state are returned.

## C TRAINING

Figs. 5, and 6 show the training curves for MDPs, and GridWorld environments, respectively, across 3 random seeds. The results in the main text correspond to the median model. We ran the experiments on Nvidia GeForce RTX 2080 Ti GPUs for context length $\leq 256$ which took approximately 12-24 hours, and on Nvidia A100 GPUs for higher context lengths, which took 1-2 days.

## D ADDITIONAL ANALYSIS

In this section, we show that $Q$-estimates, though imperfect, produce reasonable signals for task identification. Here, we test this claim thoroughly with 3 analyses.

### D.1 REQUIREMENTS FOR A UNIQUE $Q^*$-FUNCTION

Throughout, we assume fixed state space and action space. Below, we show that if the transition function is fixed, then two $Q^*$-tables will be identical if and only if both reward functions are also equal. First, we show that identical $Q^*$ functions imply identical reward functions. Given the

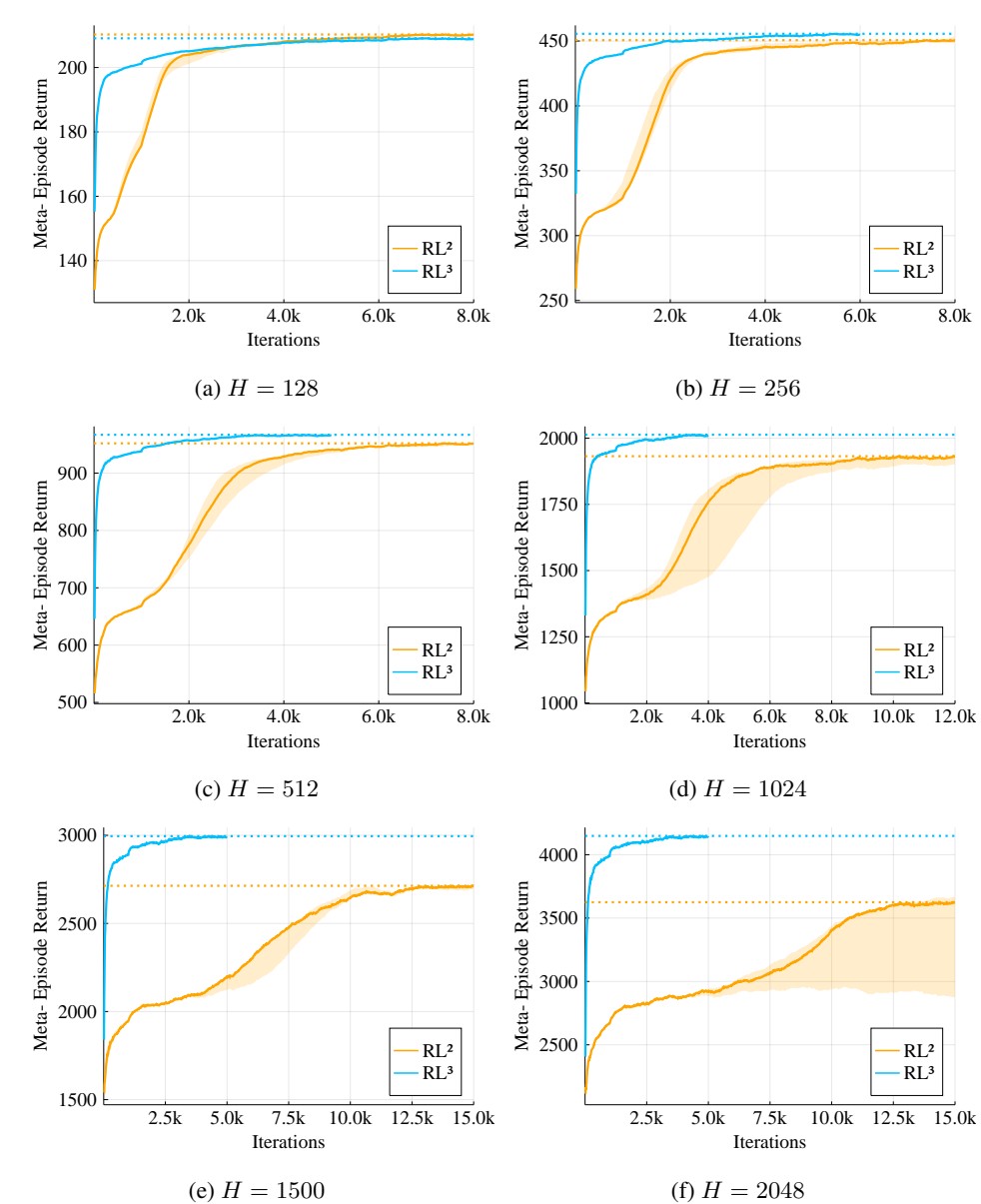

Figure 5: Average meta-episode return vs PPO iterations for MDPs domain for different interaction budgets.

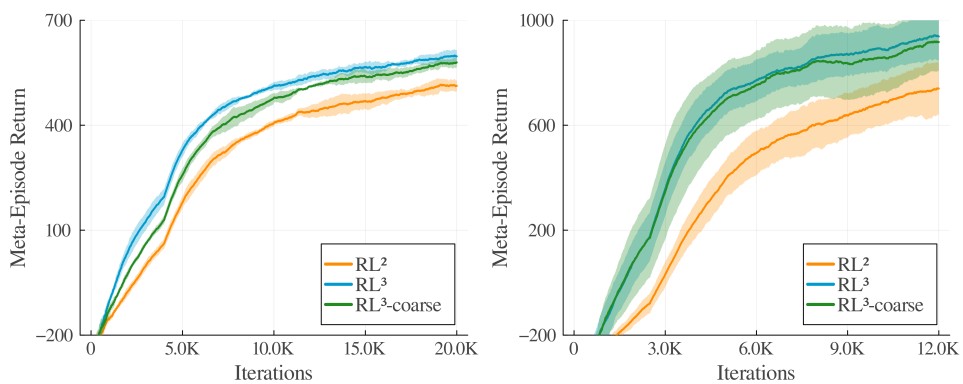

Figure 6: Average meta-episode return vs PPO iterations for GridWorld 11x11 (*left*) and 13x13 (*right*).

Bellman equations,

$$Q_1^*(s, a) = R_1(s, a) + \gamma \sum_{s'} T(s, a, s') \max_{a'} Q_1^*(s', a') \tag{20}$$

$$Q_2^*(s, a) = R_2(s, a) + \gamma \sum_{s'} T(s, a, s') \max_{a'} Q_2^*(s', a') \tag{21}$$

Substituting $Q_2^* = Q_1^*$ in Equation equation 21, we get

$$Q_1^*(s, a) = R_2(s, a) + \gamma \sum_{s'} T(s, a, s') \max_{a'} Q_1^*(s', a') \tag{22}$$

Subtracting Equation equation 20 from Equation equation 22, we get $R_1(s, a) = R_2(s, a)$. Thus, $(Q_1^* = Q_2^*) \wedge (T_1 = T_2) \implies (R_1 = R_2)$.

Now, if two MDPs have the same reward and transition function, they are the same MDP and will have the same optimal value function. So, $(R_1 = R_2) \wedge (T_1 = T_2) \implies (Q_1^* = Q_2^*)$.

Since encountering similar $Q^*$-tables is thus dependent on both transitions and rewards 'balancing' each other, the question is then for practitioners: How likely are we to get many MDPs that all appear to have very similar $Q^*$-tables?

### D.2 EMPIRICAL TEST USING MAX NORM

Given an MDP with 3 states and 2 actions, we want to find the probability that $||Q_1^* - Q_2^*||_\infty < \delta$, where $Q_1^*$ and $Q_2^*$ are 6-entry (3 states $\times$ 2 actions) $Q^*$-tables. The transition and reward functions are drawn from distributions parameterized by $\alpha$ and $\beta$, respectively. Transition probabilities are drawn from a Dirichlet distribution, $\text{Dir}(\alpha)$, and rewards are sampled from a normal distribution, $\mathcal{N}(1, \beta)$. In total, we ran 3 combinations of $\alpha$ and $\beta$, each with 50,000 MDPs, a task horizon of 10, and $\delta = 0.1$. To get the final probability, we test all $((50,000 - 1)^2)/2$ non-duplicate pairs and count the number of max norms less than $\delta$.

**Results:** For $\alpha = 1.0$, $\beta = 1.0$, we found the probability of a given pair of MDPs having duplicate $Q^*$-table to be $\epsilon = 2.6 \times 10^{-9}$. For $\alpha = 0.1$, $\beta = 1.0$, which is a more deterministic setting, we found $\epsilon = 4.6 \times 10^{-9}$. Further, with $\alpha = 0.1$, $\beta = 0.5$, where rewards are more closely distributed, we found $\epsilon = 1.1 \times 10^{-7}$. Overall, we can see that even for a set of very small MDPs, the probability of numerically mistaking one $Q^*$-table for another is vanishingly small.

### D.3 PREDICTING TASK FAMILIES

The near uniqueness of $Q^*$-functions is encouraging, but max norm is not a very sophisticated metric. Here, we test whether a very simple multi-class classifier (1 hidden layer of 64 nodes), can accurately identify individual tasks based on their *Q-estimates*. Moreover, we track how the classification accuracy improves as a function of the number of steps taken within the MDP as the estimates improve. In this experiment, the same random policy is executed in each MDP for 50 time steps. As before, our MDPs have 3 states and 2 actions.

We instantiate 10,000 MDPs whose transition and reward functions are drawn from the same distribution as before: transitions from a Dirichlet distribution with $\alpha = 0.1$ and rewards sampled from a normal distribution $N(1, 0.5)$. Thus, this is a classification problem with 10,000 classes. *A priori*, this exercise seems relatively difficult given the number of tasks and the parameters chosen for the distributions. Fig. 7 shows a compelling result given the simplicity of the model and the relative difficulty of the classification problem. Clearly, $Q$-estimates, even those built from only 20 experiences, provide a high signal-to-noise ratio w.r.t. task identification. And this is for a *random* policy. In principle, the meta-RL agent could follow a much more deliberate policy that actively disambiguates trajectories such that the $Q$-estimates evolve in a way that leads to faster or more reliable discrimination.

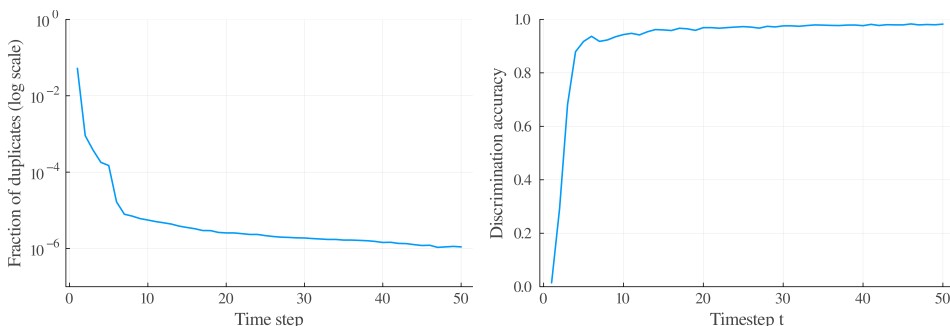

Figure 7: The task-identification power of $Q$-estimates. *Left*: Fraction of $\delta$-duplicates, with $\delta = 0.1$, as a function of time steps in a set of 5,000 random MDPs. *Right*: Accuracy of a simple multi-class classifier in predicting task ID given $Q$-table estimates, as function of time step. Both figures are generated using the same policy.

# E  DOMAIN DESCRIPTIONS

## E.1  BERNOULLI MULTI-ARMED BANDITS

We use the same setup described by Duan et al. (2016). At the beginning of each meta-episode, the success probability corresponding to each arm is sampled from a uniform distribution $\mathcal{U}(0, 1)$. To test OOD generalization, we sample success probabilities from $\mathcal{N}(0.5, 0.5)$

## E.2  RANDOM MDPS

We use the same setup described by Duan et al. (2016). The MDPs have 10 states and 5 actions. For each meta-episode, the mean rewards $R(s, a)$ and transition probabilities $T(s, a, s')$ are initialized from a normal distribution ($\mathcal{N}(1, 1)$) and a flat Dirichlet distribution ($\alpha = 1$), respectively. Moreover, when an action $a$ is performed in state $s$, a reward is sampled from $\mathcal{N}(R(s, a), 1)$. To test OOD generalization, the transition probabilities are initialized with Dirichlet $\alpha = 0.25$.

Each episode begins at state $s = 1$ and ends after `task_horizon` = 10 time steps.

## E.3  GRIDWORLDS

A set of navigation tasks in a 2D grid environment. We experiment with 11x11 (121 states) and 13x13 (169 states) grids. The agent always starts in the center of the grid and needs to navigate through obstacles to a single goal location. The goal location is always at a minimum of `min_goal_manhat` Manhattan distance from the starting tile. The grid also contains slippery wet tiles, fatally dangerous tiles and warning tiles surrounding the latter. There are `num_obstacle_sets` set of obstacles, and each obstacle set spans `obstacle_set_len` tiles, in either horizontal or vertical configuration. There are `num_water_sets` set of wet regions and each wet region always spans `water_set_length`, in either a horizontal or vertical configuration. Entering wet tiles yields an immediate reward of -2. There are `num_dangers` danger tiles and entering them ends the episode and leads to a reward of -100. Warning tiles always occur as a set of 4 tiles non-diagonally surrounding the corresponding danger tiles. Entering warning tiles causes -10 reward. Normal tiles yield a reward of -1 to incentivize the agent to reach the goal quickly. On all tiles, there is a chance of slipping sideways with a probability of 0.2, except for wet tiles, where the probability of slipping sideways is 1.

The parameters for our canonical 11x11 and 13x13 GridWorlds are: `num_obstacle_sets` = 11, `obstacle_set_len` = 3, `num_water_sets` = 5, `water_set_length` = 2, `num_dangers` = 2, and `min_goal_manhat` = 8. The parameters for the OOD variations are largely the same and the differences are as follows. For DETERMINISTIC variation, the slip probability on non-wet tiles is 0. For DENSE variation, `obstacle_set_len` is increased to 4. For WATERY variation, `num_water_sets` is increased to 8. For DANGEROUS variation, `num_dangers` is increased to

Table 3: RL$^2$/RL$^3$ Hyperparameters

| Hyperparameter | Value |
|---|---|
| Learning Rate (Actor and Critic) | 0.0003 (Bandits, MDPs) |
| | 0.0002 (GridWorlds) |
| Adam $\beta1, \beta2, \epsilon$ | $0.9, 0.999, 10^{-7}$ |
| Weight Decay (Critic Only) | $10^{-2}$ |
| Batch size | 32768 |
| Rollout Length | Interaction Budget ($H$) |
| Number of Parallel Envs | Batch Size $\div H$ |
| Minibatch Size | 4096 |
| Entropy Regularization Coeff | 0.1 with decay (MDPs) |
| | 0.04 (GridWorlds) |
| | 0.01 (Bandits) |
| PPO Iterations | See training curves |
| Epochs Per Iteration | 8 |
| Max KL Per Iteration | 0.01 |
| PPO Clip $\epsilon$ | 0.2 |
| GAE $\lambda$ | 0.3 |
| Discount Factor $\gamma$ | 0.99 |
| Decoder Layers | 2 |
| Attention Heads | 4 |
| Activation Function | gelu |
| Decoder Size ($d\_model$) | 64 |

4. For CORNER variation, `min_goal_manhat` is set to 12, so that the goal is placed on one of the corners of the grid.

There is no fixed task horizon for this domain. An episode ends when the agent reaches the goal or encounters a danger tile. In principle, an episode can last through the entire meta-episode if a terminal state is not reached.

When a new grid is initialized at the beginning of each meta-episode, we ensure that the optimal, non-discounted return within a fixed horizon of 100 steps is between 50 and 100. This is to ensure that the grid both has a solution and the solution is not trivial.

