# OpenReview forum: "RL$^3$: Boosting Meta Reinforcement Learning via RL inside RL$^2$"
_ICLR.cc/2025/Conference — Submitted to ICLR 2025_

### Official Review · Reviewer_WXJJ · 2024-10-17

**Soundness:** 2
**Presentation:** 2
**Contribution:** 2
**Rating:** 5
**Confidence:** 4

**Summary:**

Discerning that RL$^2$ shows poor asymptotic and ood performance while general RL demonstrates short-term efficiency, this paper constructively combines the general RL with the previous meta-RL algorithm RL$^2$ and calls it RL$^3$. To be specific, the paper seeks to add an extra signal as input, namely the action-value function. Though seems to be straightforward, adding this input would give a useful compression compared to trajectory history. To validate the effectiveness, the paper provides the theoretical proof for recovering the optimal meta-level value function. Finally, some experiments on discrete domains are performed.

**Strengths:**

1. The paper constructively combines the previous meta-RL algorithms called RL$^2$ and general RL signal to absorb the advantage and avoid the disadvantage of both sides.
2. The paper adopts extra action value as the auxiliary input and provides some theoretical evidence to show rationality.
3. Compared to RL$^2$, which uses the history trajectory rather than the RL$^3$ using the action value, the latter can learn more efficiently and easily. I appreciate the authors' thorough explanation.

**Weaknesses:**

1. The biggest concern for me is that as shown in Appendix A.4, $\textbf{based on the extra input action-value}$, the iteration process would lead to two cases either the belief distribution collapses rapidly to zero for tasks $j\ne i$ or the belief distribution will reduce to zero for tasks not compatible with the observed Q-values. $\textbf{Does this mean that whatever is capable of distinguishing tasks as input should meet this proof and the action value is only a feasible case?}$ If so, what is the advantage of this over adopting the task representation as input like the context-based meta-RL algorithms [1] ? Note that these algorithms can also hold a good compression w.r.t the transition / trajectory. Also, these algorithms can resolve the continuous domain rather than be limited to the discrete domain.
2. In my view, why RL$^3$ holds better OOD performance can not be interpreted by the theoretical justification.
3. RL$^2$ is a very early meta-RL algorithm, incrementally building upon RL$^2$ may still have a large gap with SOTA algorithms. As some context-based meta-RL algorithms hold the SOTA, I believe the authors can move RL$^3$ onto the context-based meta-RL algorithms to show further superiority or at least add some comparison.
4. This is related to problem 3, some lately-published papers regarding advanced meta RL should be discussed.
5. Some unclear definitions and typos: what are object-level and meta-level? What is Figure b) trying to show? Why does any permutation of transitions yield the same Q-estimates? For testing OOD generalization, MDPs are generated from distributions with different parameters than in training, what does the "parameters" mean here? How do the authors generate the OOD data? VeriBAD -> VariBAD
6. Unclear pseudo-code: Does the $ONEHOT(s)\cdot Q(s) \cdot N(s)$ denote the results after concatenation?

[1] Towards an Information Theoretic Framework of Context-Based Offline Meta-Reinforcement Learning. Lanqing Li, et.al.

$\textbf{I would be happy to raise the score if the above problems are properly addressed.}$

**Questions:**

See Weakness above.

---

> ### Author Response · Authors · 2024-11-25
> **Thank you for the review**
>
> - W1: You are right that adopting an explicit task-representation like in [1] may provide better task discrimination. In fact, Q-values may not even be equally discriminating (though extremely discriminating in practice). But the point we are making is that it is not necessary to discriminate between tasks that share the same Q-values since the optimal action is the same in each case. As Q-estimates converge, they represent just the "right amount" of belief collapse (even if not full collapse) while directly indicating the optimal action. At that point, belief can be entirely ignored for decision making since the optimal action is already indicated by Q-values, which is a simpler mapping to learn. This mechanism works even if there is a distribution shift at deploy time -- a scenario where Q-values might suggest an incorrect belief, but still suggest the correction action. By contrast, task-representation based approaches rely heavily on belief for decision making, which is not just a more complex mapping, but it also creates a single point of failure if the belief is incorrect (can happen due to distribution shift or even due to error accumulation during belief updates).
> - W2: Answered during response to W1
> - W3: While we agree with you, we emphasize that our custom implementation of RL^2, even without transformers, is known to be comparable to other state-of-the-art meta-RL baselines [2] (like VariBAD, a context-based approach), as mentioned in section 5 introduction.
> - W4: Point noted. Suggestions are welcome.
> - W5: Object-level and meta-level are terms defined in foundational work on principles of metareasoning for bounded optimality (Russell & Wefald, 1989). In our case, object-level process refers to the task-specific RL subroutine and meta-level process to the meta-RL policy. Regarding any permutation of transitions leading to the same Q-estimates, it is because in a Markovin system, the same transition model $T(s,a,s')$ and reward model $R(s,a,s')$ would be inferred regardless of the order in which the data tuples $<s,a,r,s'>$ arrive. And same $T, R$ => same MDP => same Q-values. For testing OOD generalization, the "parameters" (for procedural generation of tasks, e.g. density of obstacles or stochasticity of actions in GridWorlds) for each domain are mentioned in section 5 (and in more detail in Appendix E).
> - W6: Yes, $\cdot$ (cdot) denotes concatenation here. We have mentioned that explicitly now in the updated manuscript. Thank you for pointing it out.
>
> [2]  Recurrent model-free RL can be a strong baseline for many POMDPs. Ni et al.

---

> ### Comment · Reviewer_WXJJ · 2024-11-25
> **Response for the rebuttals**
>
> Dear authors,
>
> I appreciate the authors' clarification. The following are my responses:
>
> 1. Using Q-values to guide policy training is the core claim of this paper, but the authors' response highlights the advantages of Q-value guidance on just intuition (while I acknowledge identifying the optimal action value may be a potential advantage against task representation, this needs to be further validated through theory or experiments and it seems that the task representation can also fit in your proposed theoretical proof) rather than providing further theoretical or experimental evidence.
> Additionally, the inability of Q-value guidance to handle continuous settings is, in my view, a more significant drawback.
> I also disagree with the claim that Q-values can indicate the correct action during deployment while task representations cannot. Q-values can also fail to provide accurate estimates under distribution shifts (even their relative magnitudes may become incorrect).
> I suggest that the authors further strengthen the comparison between Q-values and representations in their submission.
>
> 2. The authors' literature review appears to focus only on works up to 2022, while many current SOTA algorithms have already surpassed VariBAD, such as [1,2].
>
> [1] Meta-Reinforcement Learning via Exploratory Task Clustering
> [2] ContraBAR: Contrastive Bayes-Adaptive Deep RL

---

> ### Author Response · Authors · 2024-12-02
> **Thank you for the review**
>
> > "I also disagree with the claim that Q-values can indicate the correct action during deployment while task representations cannot."
>
> We appreciate the suggestion that other task representations could be beneficial, but we also note that Q-values have been extensively studied and the paper provides an argument in favor of their use (e.g., Q-values provide a simpler mapping to correct actions upon convergence), while alternative representations are worth further consideration. We should also note that Q-values can always be used in conjunction with task representations.
>
> > "Q-values can also fail to provide accurate estimates under distribution shifts (even their relative magnitudes may become incorrect)."
>
> Q-values are always derived per task, without any assumptions about the task. Just like in standard RL, upon convergence, they are always correct for that task and $\text{argmax}_a Q(s,a)$ is always the best action, making distribution shift irrelevant. That said, we agree that it is possible that if the meta-RL agent does insufficient exploration initially due to distribution shift, the Q-value estimates could end up being relatively less useful. However, insufficient exploration is not a problem unique to RL$^3$.
>
> > "Q-value guidance on just intuition rather than providing further theoretical or experimental evidence."
>
> Results shown in Fig 3a and 3c provide experimental evidence that RL$^3$ performs better under distribution shift for the respective domains. The videos provided in the supplementary (most prominently "rl3-4.mp4", of which we provide snapshots in figure 4) also suggest that the agent is largely following the value function once the goal is found.
>
> > "inability of Q-value guidance to handle continuous settings is, in my view, a more significant drawback"
>
> RL$^3$ is indeed (currently) a discrete-action space meta-RL algorithm. However, several advanced approaches for extending it to continuous actions are underway and we believe they would be better presented in a separate paper. At a high level, the most basic approach involves using a simplified variation of DDPG for task-level RL and providing the Q-values of the best action and some other actions in its vicinity.
>
> Regarding citations, thank you for the pointers. We will include the following papers in our discussion of related work: 1) Towards an Information Theoretic Framework of Context-Based Offline Meta-Reinforcement Learning; 2) Meta-Reinforcement Learning via Exploratory Task Clustering; 3) ContraBAR: Contrastive Bayes-Adaptive Deep RL.

---

> ### Comment · Reviewer_WXJJ · 2024-12-03
> **Thanks for the response**
>
> Dear authors,
>
> I thank the authors for my response. Here, I'd like to restate my core concern:
>
> As said in my initial response and also seems to be agreed by the authors, the Q-value and the task representation can both meet the proof of Equation 4 in the Appendix, as the task representation would also collapse to zero for tasks j≠i, or face the condition to be identical to multiple MDPs.
>
> From this perspective, Q-value is just a special case for instantiating the proposed “proof-of-concept”, $\textbf{with the reason to choose the Q-value missing}$ (this needs experimental or theoretical evidence).
> As stated by the authors, adopting the Q-value would lead to discrete scenarios.
> Obviously, this is a disadvantage, thereby emphasizing the comparison between the Q-value and task representation to demonstrate the advantage of the Q-value.
> Despite the author's response that it deserves further consideration, it is, in my view, a significant part of the submission.
> Hence, I strongly recommend the authors compare the Q-value and the task representation to highlight their claimed edge.
>
> Additionally, I would like to restate that I remain positive about the potential of this paper.
> However, since the unique advantage of Q-value against task representation is missing, this indeed causes the paper logically incomplete, which makes me feel below the accepted bar.
> Nevertheless, I still thank the authors for resolving my other concerns, I decide to raise my score from 3->5.
>
> I hope that the authors will consider my comments and make the paper more beneficial!

---

### Official Review · Reviewer_eaSY · 2024-11-02

**Soundness:** 2
**Presentation:** 2
**Contribution:** 3
**Rating:** 5
**Confidence:** 3

**Summary:**

The paper proposes a novel hybrid approach, RL3, that integrates value-augmented MDP into meta reinforcement learning (meta-RL) frameworks RL2. The authors aim to address limitations in current meta-RL models, such as poor asymptotic performance and weak out-of-distribution (OOD) generalization. RL3 introduces object-level Q-value estimates computed using traditional RL, which are integrated into the meta-RL model to improve cumulative reward, speed up meta-training, and enhance OOD generalization.

**Strengths:**

1) RL3 may reduces meta-training time, which has potential practical implications for scaling meta-RL systems.

2) The authors provide theoretical foundation, including proofs that support the efficacy of incorporating Q-value estimates into meta-RL, explaining the potential advantages in handling OOD tasks and long-term planning.

**Weaknesses:**

1) The author’s understanding of RL2 appears to remain questionable. Specifically, the claim made in the discussion of RL3 suggests that
"For object-level RL, we use model estimation followed by value iteration (with discount factor
γ = 1) to obtain Q-estimates."
However, in reality, RL2 also employs Q-value estimation in a similar manner to predict the future expectation of reward using the discount factor γ. Specifically, both RL2 and RL3 optimize the same equation mentioned as Equation (1) in the paper. They both employ the idea of optimizing across multiple episodes simultaneously, and there does not appear to be a fundamental difference between the two approaches.

This similarity raises concerns about the novelty of the author's argument regarding RL3's distinct capabilities.

2) Authors mentioned "To test OOD generalization, we vary parameters including the stochasticity of actions, density of obstacles, and the number of dangerous tiles." Although this level of variation is commonly considered a new task in meta-learning settings, the actual changes introduced are relatively minor. Furthermore, GridWorld is a relatively simple environment.

Whether you have plans to test on more complex domains in future work, such as vision-based RL tasks or robotic control tasks (for example DMC or Atari)?

3) The authors did not provide a clear definition of the RL3 algorithm in a formalized manner, nor did they present a direct comparison with the formulation of the RL2 algorithm. This lack of precise algorithmic definitions makes it difficult for readers to fully understand how RL3 operates and differs from RL2.

Could authors  include a side-by-side comparison with RL2 to highlight the key differences.

4) I not sure that simply feeding the entire sequence of states and actions from a whole task continuously into an LSTM or transformer, as done in RL3, will necessarily enhance OOD generalization and long-term performance. A more promising approach would involve training on trajectories of both short and long lengths. Whether authors have considered alternative methods combining varying trajectory lengths.

**Questions:**

1) What is a VAMDP? Is there a clear and formal definition?

2) Could the authors outline the fundamental differences in the implementation of RL2 and RL3? Are there any precise mathematical formulations that highlight these differences? As presented, Equation 1 only expresses the meta-learning objective but does not distinguish RL2 from RL3.

3) Why is RL2 unable to address OOD generalization problems? Additionally, why does RL2, compared to RL, lack long-term performance? If RL methods have OOD generalization capabilities, why does RL2 not inherit them?

4) What is the difference between the Q-value in RL, RL2, and the Q-estimates used in RL3?

5) In Figure 3b, why are the performance results of RL2 and RL3 equal when the budget is set to 128? Which specific setting in the experiments is responsible for this outcome?

---

> ### Author Response · Authors · 2024-11-25
> **Thank you for the review**
>
> - W1: It is true that both RL$^2$ and RL$^3$ optimize the same meta-RL objective (as do most meta-RL algorithms), and that this objective considers optimizing behavior across multiple object-level episodes. We believe the confusion here lies in the fact that while many meta-RL algorithms, including RL$^2$, are described as 'performing object-level RL' within an 'inner loop', they are doing so implicitly. It is not possible to directly extract an actual object-level Q-estimate from RL$^2$. RL$^3$ on the other hand uses explicit Q-estimation (in this case, via value iteration) to produce additional inputs for the meta-agent. The VAMDP introduced later in the paper is done so in order to formalize this process.
>
> - W2: We agree that tests on more complex domains would be more convincing, although our specific hypothesis regarding the utility of object-level Q-estimates does not include SOTA performance on continuous domains as a condition. However, we do plan to extend RL$^3$ to larger/continuous state spaces, anticipating similar performance gains as the arguments made in favor of RL$^3$ would still hold.
>
> - W3: RL$^3$ differs from RL$^2$ only in that it takes an additional input: object-level Q-estimates. There are many ways one might compute these Q-estimates. We provide an example, which is to use value iteration (or value iteration on abstracted states), though this is not required.
>
> - W4: We agree that "simply feeding the entire sequence of states and actions from a whole task continuously into an LSTM or transformer" is likely not a good strategy for strong OOD or long-term performance. This is why our proposed method _augments this strategy with object-level Q-estimates_, which we argue in the paper provide (in practice) several nice properties not attainable from trajectories alone.
>
> - Q1: A VAMDP is a Value-Augmented MDP (outlined in section 4.2 and used within an algorithm in Appendix B). This construction is similar to a regular MDP except that the state space of the VAMDP includes possible Q-estimates. For example, an agent may visit the same state in the MDP multiple times, but as it performs Q-learning, its Q-estimate upon each visit will be different. This will result in a different VAMDP state.
>
> - Q2: Both RL$^2$ and RL$^3$, and indeed most meta-RL methods, solve the same objective function (Equation 1). The fundamental difference between RL$^2$ and RL$^3$ is that RL$^3$ takes an additional input (object-level Q-estimates). Importantly, _this additional input can be derived from the same input given to RL$^2$, therefore this does not constitute "extra'' or "privileged'' information_.
>
> - Q3: There is no singular reason that RL$^2$ (or any other learning algorithm) has difficulty with OOD generalization. The most common understanding is that learning algorithms are optimized to achieve high performance in-distribution by exploiting patterns specific to that data. Thus, when presented data beyond the training distribution, the new data may not follow the patterns the algorithm has learned to exploit, and the algorithm performs worse. \
> RL$^2$ seems to have worse long-term performance because the problem of mapping longer and longer (state, action, reward) trajectories to actions becomes *more difficult* over time, while RL algorithms performing Q-learning become *more accurate* over time. RL$^2$ does not, in practice, inherit the nice asymptotic properties of RL because it does not maintain object-level Q-estimates (or some equivalent) explicitly and thus it cannot benefit from this more easily exploitable representation.
>
> - Q4: In the case that you are asking about whether Q-values are different than Q-estimates: We are using these terms interchangeably. \
> In the case that you are asking for which value function the different Q-values are referring to: In RL, Q-estimates (say in DQN or in Q-learning) refer to the object-level value function of the MDP at hand, i.e. expected return-to-go within an episode of the MDP. In meta-RL (RL$^2$), the meta-value function (in this paper we use $\bar{V}$) is estimated by the meta-critic (PPO critic) during meta-training and refers to the expected return-to-go in the meta-episode. RL$^3$ computes Q-values of the MDP in a meta-episode using RL and then directs the result into the meta-RL actor and the meta-critic.
>
> - Q5: There is no specific experimental setting that causes RL$^2$ and RL$^3$ to have similar performance when the horizon is low (128). Our hypothesis is that this problem setting is relatively easy and thus RL$^2$ is able to achieve near-optimal performance. RL$^3$ likely also achieves near-optimal performance in this setting and does so in a similar number of iterations. One possible reason for the similarity in performance is that with a relatively low budget, the object-level Q-estimates within RL$^3$ have less time to converge to a useful signal and thus RL$^3$ derives less (or no) benefit from having access to them in this low-budget case.

---

### Official Review · Reviewer_5dhT · 2024-11-04

**Soundness:** 3
**Presentation:** 3
**Contribution:** 3
**Rating:** 6
**Confidence:** 3

**Summary:**

This paper presents RL3, a novel approach to meta reinforcement learning that combines traditional RL techniques with meta-RL methods. The authors propose incorporating Q-value estimates from traditional RL as additional inputs to a meta-RL architecture, aiming to improve long-term performance and out-of-distribution generalization while maintaining short-term efficiency.

**Strengths:**

- Novel approach: The paper introduces an innovative method that combines the strengths of traditional RL and meta-RL, potentially addressing some limitations of existing meta-RL approaches.
- Theoretical foundation: The authors provide theoretical insights into why incorporating Q-value estimates can be beneficial, linking them to the optimal meta-value function.
- Improved performance: RL3 demonstrates better long-term returns and out-of-distribution generalization than RL2, while maintaining short-term efficiency.
- Flexibility: The approach can be applied to various meta-RL algorithms, not just RL2.

**Weaknesses:**

- Limited baselines: The paper would benefit from comparing RL3 to more state-of-the-art meta-RL approaches beyond just RL2. In particular, comparing RL3 to hypernetwork-based approaches would provide valuable insights into its relative performance.
- Scope of experiments: The experiments are limited to discrete domains. As suggested, it would be valuable to see how RL3 performs in high-dimensional state spaces and with continuous action spaces. This limitation restricts the applicability of the current results to more complex, real-world scenarios.
- Compute budget dependency: The performance of RL3 relative to RL2 seems to depend on the compute budget (H), which appears to be environment-dependent. The paper would benefit from providing guidelines or heuristics for selecting an appropriate range of H values for different environments.
- Generalizability: While the paper demonstrates improved out-of-distribution generalization, it's unclear how well this generalizes across a wider range of task distributions or more complex domains.
- Computational overhead: The paper doesn't thoroughly discuss the potential computational overhead of computing Q-value estimates alongside the meta-RL process, which could be a significant factor in practical applications with high dimensional state spaces.

**Questions:**

See Weakness.

---

> ### Author Response · Authors · 2024-11-25
> **Thank you for the review**
>
> - W1: While we agree with you, we emphasize that our custom implementation of RL$^2$, even without transformers, is known to be comparable to other state-of-the-art meta-RL baselines [2], as mentioned in section 5 introduction.
> - W2: We agree with your assessment. This paper is a proof of concept (as mentioned in the introduction), and our specific hypothesis does not include SOTA performance on continuous domains as a condition. However, we do plan to extend RL$^3$ to larger/continuous state spaces, anticipating similar performance gains as the arguments made in favor of RL$^3$ would still hold.
> - W3: Interaction budget (or adaptation period) $H$ is part of the problem specification. It is not a hyperparameter. RL$^3$'s advantage over RL$^2$ increases with $H$.
> - W5: Regarding computation overhead for scaling to larger state spaces, several strategies can be employed, like state-abstractions to reduce state space size (which we show leads to substantial computational savings in gridworlds), or using a more-efficient planner than value iteration (e.g., RTDP or MCTS). In fact, it is even possible to do value iteration on a GPU, which would compute Bellman updates for all states at once [3]. For continuous state spaces, we speculate that fitting Q-values to even a small function approximator would be sufficient to benefit RL$^3$.
>
> [2] Recurrent model-free RL can be a strong baseline for many POMDPs. Ni et al. \
> [3] Markov Decision Process Parallel Value Iteration Algorithm On GPU. Chen et al. 2013.

---

> > ### Comment · Reviewer_5dhT · 2024-11-26
> >
> > Thank you for your detailed responses and clarifications. We appreciate the insights you provided regarding the implementation and future directions of your work. Here are my thoughts:
> > - Performance in Continuous Action Spaces and Larger State Spaces: While I understand that this paper serves as a proof of concept, demonstrating the algorithm's effectiveness in continuous action spaces and larger state spaces would significantly strengthen its contributions. Such capabilities are crucial for broader applicability and would enhance the impact of your work.
> > - Baselines I noticed that the cited paper is from 2021. Given the advancements in the field, it would be interesting to see how your approach compares to more recent developments, such as hypernetworks[1][2]. These methods have shown promise in meta-reinforcement learning and could provide valuable context for evaluating your algorithm's performance.
> > I look forward to seeing how these considerations might be integrated into future iterations of your research. Thank you again for your engagement and the opportunity to discuss these aspects of your work.
> >
> >
> > [1] Hypernetworks in Meta-Reinforcement Learning. Jacob Beck, Matthew Thomas Jackson, Risto Vuorio, Shimon Whiteson
> >
> > [2] Recurrent Hypernetworks are Surprisingly Strong in Meta-RL. Jacob Beck, Risto Vuorio, Zheng Xiong, Shimon Whiteson

---

### Official Review · Reviewer_E1NK · 2024-11-06

**Soundness:** 2
**Presentation:** 4
**Contribution:** 2
**Rating:** 5
**Confidence:** 4

**Summary:**

The paper presents a modification of meta-RL that incorporates additional information in the form of Q-estimates and state counts (and possibly other things as well) in order to improve training times, data requirements and possibly asymptomatic performance.  The approach is motivated with both empirical and theoretical arguments and validated on three artificial test domains.

**Strengths:**

1. The paper is *very* well-presented.  It was easy to understand and enjoyable to read.
2. The work is well-situated in the literature.
3. The paper provides both empirical and theoretical analyses that both motivate and justify the approach and design decisions.
4. Reported results are good and convince the reader that the proposed approach does provide (some of) the improvements hypothesized.  In particular, rewards are comparable or better than RL^2 while (meta)training time is decreased and OOD generalization seems also to be improved to some extent.

**Weaknesses:**

1. Experimental results are obtained on only toy problems.  It seems likely that the approach may not scale to more difficult/larger/real-world problems.  How does this work on a modestly more difficult domain like Atari, for example?
2. Some claims are not really supported.  For example, while the following statement from Sec. 5 seems like it could be true, there is no supporting evidence given: "VAMDPs can be plugged into any base meta-RL algorithm with a reasonable expectation of improving it."  As another example, it is also suggested that RL^3 might enjoy convergence guarantees, but this is also not supported in any way.
3. Performance improvement results seem like they don't include the overhead for computing q-values and state counts and are therefore potentially somewhat misleading.  This is addressed to some degree in the Sec 6, "Computation Overhead Considerations" subsection, but the discussion is too brief to be convincing; in particular, it is not clear that the claim will hold if scaled to larger/real-world problems.
4. The paper doesn't compare RL^3 with any other meta-RL approaches, other than RL^2, the one it modifies.

**Questions:**

1. Several places in the paper seem to suggest that one of the benefits of RL^3 will be asymptotic guarantees of convergence.  Does RL^3 in fact, gain asymptotic performance guarantees?

2. In Section 4.2 it states, "In practice, we provide action advantages along with the max Q-value (value function) and optionally, standard errors, instead of just Q-estimates".  How critical is this to RL^3's success?  How much does this affect the results?

3. Sec 5 talks about using PPO as the metalearner, but earlier the paper talks about using a blackbox/NN for the metalearner.  I'm a bit confused here---is PPO used just for augmented RL^2?

4. Is Fig 3b comparing only meta-training time (as the last sentence of the MDP results subsection seems to suggest)? Or does it include all required computation?  How are the q-values and counts obtained? How much does this cost? How would this scale for larger/real-world problems?

---

> ### Author Response · Authors · 2024-11-25
> **Thank you for the review**
>
> - W1: We agree with your assessment. This paper is a proof of concept (as mentioned in the introduction), and our specific hypothesis does not include SOTA performance on continuous domains as a condition. However, we do plan to extend RL^3 to larger/continuous state spaces, anticipating similar performance gains as the arguments made in favor of RL^3 would still hold.
> - W2: Our expectation (but not a definitive claim) that plugging in VAMDPs to an off-the-shelf base meta-RL algorithm could enhance its performance is based on the observation that the arguments made in favor of RL$^3$ are agnostic to the base meta-RL algorithm. Regarding convergence guarantees, we do not claim any guarantees. However, it is reasonable to expect the task-specific RL component to converge to near-optimal Q-values in the limit (subject to sufficient exploration), and RL$^3$ to be able to learn (subject to deep learning limitations) the trivial mapping from optimal Q-values to greedy actions.
> - W3: See response to Q4.
> - W4: While we agree with you, we emphasize that our custom implementation of RL$^2$, even without transformers, is known to be comparable to other state-of-the-art meta-RL baselines [2], as mentioned in section 5 introduction.
>
> - Q1: See our response to W2.
> - Q2: Thank you for asking this question. While re-doing some experiments to answer your question, we discovered that standard-errors are no longer necessary with the final optimized hyperparameters. In our earlier experiments, before the hyperparameters were fully optimized, we did observe significant benefits from including standard errors. Regarding including Q-values vs. including action advantages, while they carry the same information, action advantages are smaller in scale and therefore friendlier for neural networks. In the updated manuscript, we have removed the reference to standard-errors.
> - Q3: PPO is used for meta-training the meta-RL policy. The meta-RL policy is a blackbox recurrent model (in our implementation of RL$^2$, just a transformer mapping experience-sequences to actions). To augment RL$^2$ inputs with Q-estimates, value-iteration (over an estimated model of the environment) is used.
> - Q4: The Fig. 3b y-axis compares the number of PPO _iterations_ in meta-training, not wall-time. However, they are highly correlated as the computation time in meta-training is bottlenecked by gradient updates. The exact wall-time overhead of computing q-values (which is done using value iteration over an estimated model of the task-specific MDP) is discussed in the subsection "Computation Overhead Considerations". For scaling to larger state spaces, several strategies can be employed, including state-abstractions to reduce state space size (which we show leads to substantial computational savings in gridworlds), or using a more efficient planner than value iteration (e.g., RTDP or MCTS). In fact, it is even possible to do value iteration on a GPU, which would compute Bellman updates for many states at once [3].
>
> Once again, thank you for the review. It has helped us improve our manuscript.
>
> [2] Recurrent model-free RL can be a strong baseline for many POMDPs. Ni et al. \
> [3] Markov Decision Process Parallel Value Iteration Algorithm On GPU. Chen et al. 2013.

---

> > ### Comment · Reviewer_E1NK · 2024-11-26
> > **Responding to rebuttals**
> >
> > Thanks for answering my questions and responding to my review.  It looks like we agree that this paper has good potential but is at this point still somewhat preliminary for inclusion in a venue like ICLR.  As is also pointed out by other reviewers, further development of the work that incorporates continuous state spaces and better elucidates/demonstrates the real benefits of the proposed method will make the paper much stronger.

---

> > > ### Author Response · Authors · 2024-12-02
> > > **Thank you for the follow up**
> > >
> > > We appreciate your follow up comments. While RL$^3$ is currently a discrete-action space meta-RL algorithm, several advanced approaches for extending it to continuous domains are underway and we believe they would be better presented in a separate paper. To give you a preview, one of the most basic approaches for handling continuous actions involves using a simplified variation of DDPG for task-level RL and providing the Q-values of the best action and some other actions in its vicinity.

---

### Official Review · Reviewer_Lu7E · 2024-11-08

**Soundness:** 3
**Presentation:** 3
**Contribution:** 3
**Rating:** 5
**Confidence:** 5

**Summary:**

This paper proposes RL^{3}, a novel method designed to enhance out-of-distribution (OOD) generalization and reduce meta-training time in meta-reinforcement learning (meta-RL) by integrating Q-estimates as additional input information for the meta-RL agent. The approach is implemented within the RL^{2} framework using transformers and is evaluated against RL^{2} across several benchmark tasks, including bandits, random MDPs, and gridworld.

**Strengths:**

- By adding Q-estimates, RL^{3} demonstrates improved efficiency over RL^{2} by requiring fewer PPO iterations and delivering better OOD performance on the selected benchmarks.

**Weaknesses:**

- The experimental benchmarks focus on tasks with limited state spaces, which may restrict the applicability of the findings.
- Additional experiments on benchmarks with continuous state spaces, such as parameterized MuJoCo environments, are necessary to fully evaluate RL^{3}’s practical benefits over RL^{2} in terms of sample efficiency. The results based on tabular Q-function estimation in small state spaces do not directly indicate performance gains in more complex, real-world settings.

**Questions:**

1. Could cumulative reward plots over timesteps for each method be added to visualize meta-training progress?
2. What rollout length was used for RL^{2} in the experiments?

---

> ### Author Response · Authors · 2024-11-25
> **Thank you for the review**
>
> We agree with your assessment of the weaknesses. This paper is a proof of concept (as mentioned in the introduction), and our specific hypothesis does not include SOTA performance on continuous domains as a condition. However, we do plan to extend RL^3 to larger/continuous state spaces, anticipating similar performance gains as the arguments made in favor of RL^3 would still hold.
> Regarding your questions:
> 1. We have added the meta-training plots to the appendix.
> 2. The context/rollout length for RL$^2$ (and RL$^3$) transformer is $H$ i.e., the full history of experiences in the meta-episode.

---

> > ### Comment · Reviewer_Lu7E · 2024-12-02
> >
> > I appreciate your approach to incorporating Q-value estimates into RL$^{2}$ to enhance OOD generalization. While the results demonstrate promise, the evaluation appears to be limited to environments with a relatively small number of states. To strengthen the analysis, I suggest exploring the implications of Q-value estimates when leveraging function approximators, as opposed to tabular Q-functions, to better understand their scalability and robustness.
> >
> > Expanding the evaluation protocol to include more diverse and complex environments, such as MuJoCo, would offer additional insights and provide a stronger foundation for your findings.
> >
> > Your proof-of-concept results are encouraging, and I look forward to seeing a more comprehensive and detailed iteration of this work in a future conference.

---

### Meta-Review · Area_Chair_PZ2p · 2024-12-19

**Metareview:**

The authors propose a meta-RL framework based on prior work RL^2 that incorporates Q-estimates for better sample efficiency, asymptotic performance, and OOD generalization. They instantiate this algorithm as a modification of RL^2 with transformers and compare against RL^2.

Reviewers found the paper very well written and clear, properly situated within the related literature. There were also both empirical and theoretical analyses to motivate and justify the approach and design. The empirical results are significant compared to the incorporated baselines. However, reviewers also found that those empirical results were only evaluated in toy environments and some competitive baselines were missing. Further, some claims were not supported, including generalization claims given the limitations of the environments used.

While the idea, motivation, and algorithm are interesting, I agree with reviewers that more compelling justification is needed by incorporating results from more complex environments and compared with more recent meta-RL baselines. For these reasons, I vote to reject this paper in the current form, and encourage the authors to revise and re-submit.

**Additional Comments On Reviewer Discussion:**

In the reviewer discussion, multiple reviewers expressed concern about lack of baselines and complex environments, which were insufficiently addressed during the rebuttal phase.

---

### Decision · Program_Chairs · 2025-01-22

Reject